 Select

# On a class of selection rules without group actions in field theory and string theory

Justin Kaidi[1], Yuji Tachikawa[2] and Hao Y. Zhang[2]

**1** Department of Physics, University of Washington, Seattle, WA, 98195, USA
**2** Kavli Institute for the Physics and Mathematics of the Universe (WPI),
University of Tokyo, Kashiwa, Chiba 277-8583, Japan

## Abstract

We discuss a class of selection rules which i) do not come from group actions on fields, ii) are exact at tree level in perturbation theory, iii) are increasingly violated as the loop order is raised, and iv) eventually reduce to selection rules associated with an ordinary group symmetry. We start from basic field-theoretical examples in which fields are labeled by conjugacy classes rather than representations of a group, and discuss generalizations using fusion algebras or hypergroups. We also discuss how such selection rules arise naturally in string theory, such as for non-Abelian orbifolds or other cases with non-invertible worldsheet symmetries.

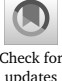

# 1  Introduction and summary

Selection rules are the most elementary manifestation of a group-like symmetry in any quantum system, including in quantum field theory. Indeed, there are results in algebraic quantum field theory which effectively say that any conservation law which holds non-perturbatively in four or more spacetime dimensions comes from group actions on fields [1]. There are, however, many known "selection rules" which are not directly associated with group-like symmetries. These are obeyed at low orders in perturbation theory, but are violated at higher loop order.[1]

In this paper, we introduce a general class of selection rules in quantum field theories which do not come from group actions on fields. These selection rules arose from the authors' study of the spacetime manifestation of non-invertible symmetries on the worldsheet in perturbative string theory. Without further ado, let us describe our selection rules.

To discuss ordinary selection rules for a symmetry described by a group $G$, each field $\phi_i$ is labelled by a particular representation $R_i$ of $G$, such that $(\phi_i)^*$ belongs to $\overline{R_i}$. Then a process involving incoming fields $\phi_{1,2,\dots,n}$ and outgoing fields $\phi_{n+1,n+2,\dots,m}$ is non-vanishing only when

$$\text{id} \subset \overline{R_1 R_2} \cdots \overline{R_n} R_{n+1} R_{n+2} \cdots R_m \,, \tag{1}$$

where id is the identity representation.

The simplest examples of our more general selection rules still involve a group $G$, but now each field $\phi_i$ is labeled by a *conjugacy class* $[g_i]$ of $G$, such that $(\phi_i)^*$ belongs to $\overline{[g_i]} = [g_i^{-1}]$. We will then be interested in theories for which every interaction in the Lagrangian $\phi_1 \dots \phi_n \subset \mathcal{L}$ satifies $\tilde{g}_1 \dots \tilde{g}_n = e$ for some $\tilde{g}_i \in [g_i]$, where $e \in G$ is the identity. As we will see below, this constraint on the Lagrangian extends to a constraint on *tree-level* processes. In particular, a tree-level process involving incoming fields $\phi_{1,2,\dots,n}$ and outgoing fields $\phi_{n+1,n+2,\dots,m}$ is non-vanishing only when we have

$$\tilde{g}_1^{-1} \tilde{g}_2^{-1} \cdots \tilde{g}_n^{-1} \tilde{g}_{n+1} \tilde{g}_{n+2} \cdots \tilde{g}_m = e \,, \tag{2}$$

for some suitably chosen $\tilde{g}_i \in [g_i]$. This selection rule holds for arbitrary number of external fields at tree-level. However, it is violated at loop order (see (13)), and at sufficiently high loop order it reduces to standard selection rules coming from a symmetry group $\text{Ab}[G] := G/[G,G]$, the abelianization of $G$.

---

[1] Readers might recall the restrictions on allowed helicities in gluon scattering [2,3] or the two-loop vanishing of the electron dipole moment in the Standard Model [4–6].

More generally, our selection rules will involve a fusion algebra $\mathcal{A}$ with fusion rules $ab = \sum_c N_{ab}^c c$. We label our fields $\phi_i$ by elements $a_i$ of this algebra, and demand that all terms appearing in the Lagrangian are consistent with the fusion rules, namely that $e \prec a_1 \ldots a_n$, meaning that the coefficient of the identity $e$ in the decomposition of the right-hand side is non-zero. As before, this constraint extends to a constraint on *tree-level* processes. Indeed, labelling the incoming fields by $a_1, \ldots, a_n$ and the outgoing fields by $a_{n+1}, \ldots, a_m$, the non-vanishing tree-level processes will be seen to satisfy

$$e \prec \overline{a_1 a_2} \cdots \overline{a_n} a_{n+1} a_{n+2} \cdots a_m \,. \tag{3}$$

This selection rule holds at tree-level for arbitrary number of external fields, but is increasingly violated at higher loop order (see (36)), eventually stabilizing to the selection rules coming from a certain Abelian group, to be specified below. This class of examples includes as a subset the ones described in the previous paragraph, where we make use of the subalgebra of the group algebra generated by conjugacy classes.

Some string theorists will immediately recognize our first example (2) as the selection rules for an orbifold by a non-Abelian group $G$; in the modern language, the worldsheet theory of a non-Abelian orbifold by $G$ has a non-invertible symmetry given by Rep$(G)$.[2] Our general case corresponds to the spacetime selection rules coming from a more general non-invertible symmetry on the worldsheet.

The rest of this paper is organized as follows. In Section 2, we introduce and discuss the details of our selection rules, first by studying selection rules dictated by conjugacy classes of groups, and then by generalizing to the case of more general fusion algebras. In Section 3, we present various concrete examples of our selection rules in the context of perturbative string theory, including non-Abelian orbifolds, worldsheet theories having Ising symmetry, strings propagating on $S^1/\mathbb{Z}_2$, and worldsheet theories having TY($\mathbb{Z}_3$) symmetry.

We also provide three appendices. In Appendix A we provide a review of the theory of finite hypergroups, which slightly generalize fusion algebras. This material is not new, but is included for the reader's convenience. In Appendix B, we give a detailed discussion of the selection rules for fields whose fusion rules are dictated by a WZW fusion category. Finally, in Appendix C we give examples of low-rank fusion algebras whose selection rules are (partially) preserved at one-loop, and are completely broken only at higher-loop order.

**Note added:** While this paper was nearing completion, we learned of another work [8] that explored the spacetime interpretation of non-invertible symmetries on the string worldsheet.

## 2 Basics

### 2.1 From conjugacy classes of groups

**Assumptions:** Let us begin by studying selection rules based on conjugacy classes of groups. Fix a finite group $G$, and consider a perturbative quantum field theory where each field $\phi_i$ is labeled by a conjugacy class $[g_i]$, a particular element of which is $g_i \in G$. We assume that the conjugate field $(\phi_i)^*$ is labeled by the class $[g_i^{-1}]$. We use this operation to regard every incoming line labeled by $[g_i]$ as an outgoing line labeled by $[g_i^{-1}]$.

Our fundamental assumption is that every bare interaction term in the Lagrangian,

$$O = \phi_1 \phi_2 \cdots \phi_n \,, \tag{4}$$

---

[2]Moreover, the fact that such selection rules can get modified beyond tree level was already mentioned in a footnote of one of the original papers about interactions on orbifolds [7].

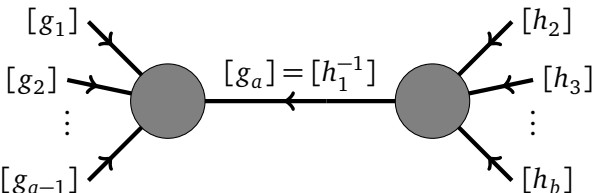

Figure 1: The $(k+1)$-vertex $(a+b-2)$-point diagram appearing in our inductive argument.

satisfies the condition that there exists $\tilde{g}_i \in [g_i]$ such that

$$\tilde{g}_1 \cdots \tilde{g}_n = e\,. \tag{5}$$

Here and below, we omit Lorentz indices or spacetime derivatives, which play no role in this paper. We also note that the above condition does not depend on the ordering of fields $\phi_1$, ..., $\phi_n$ within $O$, since

$$\tilde{g}_i \tilde{g}_{i+1} = \tilde{g}_{i+1}(\tilde{g}_{i+1}^{-1} \tilde{g}_i \tilde{g}_{i+1}) = \tilde{g}_{i+1} \hat{g}_i\,, \tag{6}$$

where once again $\hat{g}_i \in [g_i]$.

**When $G$ is Abelian:** Note that when $G$ is Abelian, each conjugacy class contains a single element, and therefore labeling fields by conjugacy classes is the same as labeling fields by group elements. Therefore, in that case, we simply have a theory whose symmetry is $G$ (or if one is more mathematically inclined, the Pontryagin dual $\hat{G}$ of $G$). Such selection rules are well-known to be preserved at all-loop order. This means that our main interest lies in the case when $G$ is non-Abelian.

**Tree-level properties:** We begin by showing that any nonzero tree diagram generated from these bare interaction terms must satisfy the same condition (5), where each $\tilde{g}_i$ is associated to an external leg of the diagram. We prove this by means of mathematical induction on the number of vertices.

When there is only one vertex, there is nothing to prove. So let us assume that we have shown the property to $k$ vertices. Now, any tree diagram with $k+1$ vertices can be cut into two tree diagrams with less than $k+1$ vertices. Let us say that one part of the diagram has external lines labeled by $[g_1], \ldots, [g_a]$ and the second part of the diagram has those labeled by $[h_1], \ldots, [h_b]$. By the inductive hypothesis, we have

$$\tilde{g}_1 \tilde{g}_2 \cdots \tilde{g}_a = e\,, \qquad \tilde{h}_1 \cdots \tilde{h}_b = e\,, \tag{7}$$

where $\tilde{g}_i \in [g_i]$ and $\tilde{h}_i \in [h_i]$. Let us say that the two external lines labeled by $[g_a]$ and $[h_1]$ arose from the cut, so that $g_a^{-1} \in [h_1]$, c.f. Figure 1. Then $\tilde{g}_a^{-1} \in [\tilde{h}_1]$, and so there is an $x \in G$ such that $\tilde{g}_a^{-1} = x\tilde{h}_1 x^{-1}$. Now define $\hat{h}_i := x\tilde{h}_i x^{-1} \sim h_i \in [h_i]$, which satisfy

$$\hat{h}_1 \cdots \hat{h}_b = e\,. \tag{8}$$

We then have

$$\tilde{g}_1 \cdots \tilde{g}_{a-1} \hat{h}_2 \cdots \hat{h}_b = (\tilde{g}_1 \cdots \tilde{g}_{a-1} \tilde{g}_a)(\hat{h}_1 \hat{h}_2 \cdots \hat{h}_b) = e\,, \tag{9}$$

which is what we wanted to prove.

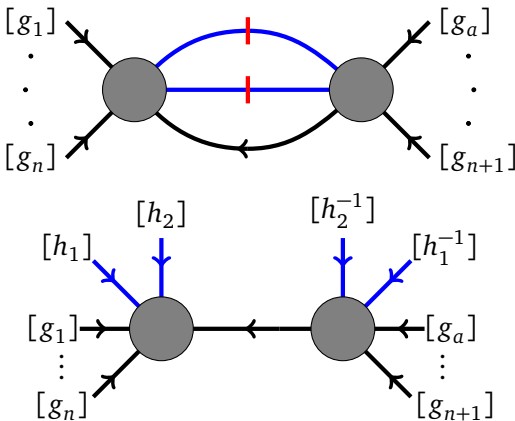

Figure 2: An example of a two loop diagram with external legs labelled by $[g_1], \ldots, [g_a]$ (left). Upon cutting the diagram in two places (red), we get a connected tree diagram with four additional external legs labelled by $[h_i]$ and $[h_i^{-1}]$ for $i = 1, 2$ (right).

**Loop order:** When we have loops in the diagram, the selection rule no longer holds in the form given above. Consider an $L$-loop diagram with $a$ external lines labeled by $[g_1], \ldots, [g_a]$. We can cut it in $L$ places to obtain a connected tree diagram with $2L$ additional external lines, labeled by $[h_1], [h_1^{-1}], \ldots, [h_L], [h_L^{-1}]$; for a concrete example, see Figure 2. We have already shown that there exist $\tilde{g}_i \in [g_i]$, $\tilde{h}_j \in [h_j]$, and $\tilde{k}_j^{-1} \in [h_j^{-1}]$ such that

$$(\tilde{g}_1 \cdots \tilde{g}_a)(\tilde{h}_1 \tilde{k}_1^{-1}) \cdots (\tilde{h}_L \tilde{k}_L^{-1}) = e, \tag{10}$$

or equivalently,

$$\tilde{g}_1 \cdots \tilde{g}_a = (\tilde{k}_L \tilde{h}_L^{-1}) \cdots (\tilde{k}_1 \tilde{h}_1^{-1}). \tag{11}$$

We now note that there exists an $x_j \in G$ such that $\tilde{k}_j = x_j \tilde{h}_j x_j^{-1}$. Then $\tilde{k}_j \tilde{h}_j^{-1} = x_j \tilde{h}_j x_j^{-1} \tilde{h}_j^{-1} = [x_j, \tilde{h}_j]$ is the commutator of two elements $x_j$ and $\tilde{h}_j$. Denoting by $\text{Com}(G)$ the set of all commutators of $G$ and by $\text{Com}(G)^L$ the set of products of $L$ commutators, we thus find that for an $L$-loop diagram,

$$\tilde{g}_1 \cdots \tilde{g}_a \in \text{Com}(G)^L. \tag{12}$$

In other words, we have the following result,

*The $L$-loop scattering amplitude with $n$ legs labeled by $[g_1], \ldots, [g_n]$ is non-zero only when*

$$\tilde{g}_1 \cdots \tilde{g}_n \in \text{Com}(G)^L, \quad \text{for some} \quad \tilde{g}_i \in [g_i]. \tag{13}$$

This is clearly less restrictive than the tree-level selection rule, which required that $\tilde{g}_1 \cdots \tilde{g}_n = e$ for some $\tilde{g}_i \in [g_i]$.

For large enough $L$, $\text{Com}(G)^L$ stabilizes and becomes the commutator subgroup $[G, G] \subset G$. As a result, for sufficiently large $L$, the condition above reduces to

$$\tilde{g}_1 \cdots \tilde{g}_a \in [G, G], \tag{14}$$

or equivalently,

$$\underline{g_1} \cdots \underline{g_a} = \underline{e} \in \text{Ab}[G] := G/[G, G], \tag{15}$$

where $\mathrm{Ab}[G] = G/[G, G]$ is the abelianization of $G$ and $\underline{g}$ is the projection of $g \in G$ to $\mathrm{Ab}[G] = G/[G, G]$. This means that, upon accounting for arbitrarily high loop orders, the selection rules obtained are equivalent to those coming from the Abelian group $\mathrm{Ab}[G] = G/[G, G]$.[3]

**Commutator length, i.e. at which loop order do our selection rules reduce to ordinary ones:** A natural question which arises is the following: given a non-Abelian finite group $G$, at which loop order $L$ do our selection rules based on conjugacy classes of $G$ reduce to ordinary selection rules based on the Abelian group $\mathrm{Ab}[G] = G/[G, G]$? Mathematically, this is equivalent to asking for the smallest $L$ for which $\mathrm{Com}(G)^L = [G, G]$. Such an $L$ is known as the *commutator length* of the finite group $G$.[4]

Let us denote the commutator length of a group by $\mathrm{cl}(G)$. Note that $\mathrm{cl}(G) = 1$ is equivalent to $[G, G] = \mathrm{Com}(G)$, i.e. the set of commutators forms a subgroup. By scanning over non-Abelian finite groups with increasingly large $|G|$, it has been found that the smallest groups with $\mathrm{cl}(G) = 2$ are two non-isomorphic groups with $|G| = 96$, see [11]. That simple non-Abelian finite groups $G$ always have $\mathrm{cl}(G) = 1$ is the famous conjecture of Ore,[5] which was proven using the classification of finite simple groups [12]. In general, it is known that $\left|[G, G]\right| \geq (\mathrm{cl}(G) + 1)!(\mathrm{cl}(G) - 1)!$, as shown in [13]. Furthermore, for any integer $L$, it is known that there is a $G$ for which $\mathrm{cl}(G) \geq L$.[6]

## 2.2 String theory realization: Non-Abelian orbifolds

Let us recall that the structure described above, coming from the conjugacy classes of a finite group $G$, naturally arises when we consider a non-Abelian orbifold by $G$ in string theory [7]. The point is that in perturbative string theory on a non-Abelian orbifold $M/G$, each twisted sector is labeled by a conjugacy class $[g]$ for $g \in G$. A tree-level amplitude with $n$ insertions is then nonzero only when there is a consistent worldsheet configuration on the sphere with insertions of $[g_1], \ldots, [g_n]$. This requires us to have elements $\tilde{g}_i \sim g_i$ such that

$$\tilde{g}_1 \tilde{g}_2 \cdots \tilde{g}_n = e, \tag{16}$$

which is exactly the condition encountered above.

At $L$ loop order, the worldsheet is a genus-$L$ surface, and the holonomy on the worldsheet is required to satisfy

$$\tilde{g}_1 \tilde{g}_2 \cdots \tilde{g}_n = [x_1, y_1][x_2, y_2] \cdots [x_L, y_L], \tag{17}$$

where $x_i, y_i$ are the holonomies around the $i$-th A-cycle and $i$-th B-cycle, respectively. Again, this is exactly the structure we found in (11). In Section 3 we will illustrate these results more explicitly by means of some concrete examples of non-Abelian orbifolds.

---

[3] A very similar result in a slightly different context can be found in [9].

[4] Commutator lengths of individual elements of $[G, G]$ can be similarly defined. The asymptotic behavior of the commutator lengths of elements of infinite discrete groups is an active area of mathematical research, see e.g. [10]. The authors thank the tweet https://twitter.com/skein_relation/status/1717954828301996053 for this information.

[5] Apparently the conjecture does not really go back to Ore, see https://mathoverflow.net/questions/77398/.

[6] This math stackexchange answer https://math.stackexchange.com/a/7885 by Derek Holt constructs an explicit example as follows. Fix a prime $p$, and consider a group $G$ generated by $a_i$ for $1 \leq i \leq n$ and $b_{ij}$ for $1 \leq i < j \leq n$, all of order $p$, with the relation $[a_i, a_j] = b_{ij}$ where $b_{ij}$ are all central. The group $G$ satisfies $|G| = p^{n(n+1)/2}$ and $\left|[G, G]\right| = p^{n(n-1)/2}$. Now note that $[ax, by] = [a, b]$ for $x, y$ in the center of $G$. Therefore, $\mathrm{Com}(G)$ contains at most $(p^n)^2$ elements, and so $\mathrm{cl}(G) \geq (n-1)/4$.

## 2.3  Using fusion algebras

We now generalize the construction in the last subsection to more general fusion algebras.[7]

### 2.3.1  The setting

**Assumptions:**  Consider an algebra $\mathcal{A}$ with a finite set $A = \{e, x, y, \ldots\}$ of basis elements,[8] with an associative multiplication law

$$x y = \sum_{z \in A} N_{xy}^z z, \tag{18}$$

with $e$ as the unit. Each element $a \in \mathcal{A}$ can be expanded in terms of basis elements $x \in A$ as

$$a = \sum_{x \in A} a_x x, \tag{19}$$

and we use the notation $x \prec a$ to indicate that the expansion coefficient $a_x$ is positive. More generally, we define $0 \prec a$ to mean that $0 \leq a_x$ for all $x$, and $b \prec a$ to mean $0 \prec a - b$.

We will assume the existence of an involution $A \to A$, which we denote by $a \mapsto \overline{a}$, with the condition that, for simple elements $x, y$, we have $e \prec xy$ (i.e. $N_{xy}^e \neq 0$) if and only if $y = \overline{x}$. To be slightly more general, we may demand that only $x$, but not $b$, is a simple element, in which case we have the property that $e \prec xb$ if and only if $\overline{x} \prec b$.

In explicit examples, the structure constants $N_{xy}^z$ are often non-negative integers, motivating one to call such a structure a *fusion algebra*. For the developments in this paper though, we will not actually need the condition $N_{xy}^z \in \mathbb{Z}_{\geq 0}$, but rather only that $N_{xy}^z \geq 0$. In these cases the structure in question is more often called a *hypergroup*, as originally introduced in [14].[9] For the reader's convenience, we have included a short Appendix A providing an overview of the basic theory of finite hypergroups. A simple consequence of $N_{xy}^z \geq 0$ is that if $a \prec b$ and $c \prec d$, we have $ac \prec bd$.

We now consider a perturbative quantum field theory whose fields $\phi_i$ are labeled by basis elements $x_i \in A$. We assume that the conjugate fields $(\phi_i)^*$ are labeled by $\overline{x_i}$. We further assume that the bare interaction terms

$$O = \phi_1 \phi_2 \cdots \phi_n, \tag{20}$$

satisfy the condition

$$e \prec x_1 x_2 \cdots x_n. \tag{21}$$

Here, the ordering of the fields within an interaction in (20) is not usually meaningful. As such, we assume that the fusion algebra $A$ controlling the condition (21) is commutative in the rest of the paper.[10]

---

[7]Fusion algebras are sufficient to describe the scattering of point-like operators. If one wants to generalize to scattering between objects of different dimensionalities, then one presumably needs to use a higher fusion algebra instead.

[8]We remark that, as long as there are a finite number of fields in the Lagrangian, even if $A$ (e.g. $\mathrm{Rep}(G)$) were infinite, we would still only use a finite number of elements as labels.

[9]The term *hypergroup* can refer to a weaker structure depending on the literature, where one considers the multiplication as a map $m : A \times A \to \{S \mid S \subset A\}$ satisfying a certain associativity condition. A hypergroup in this weaker sense can be obtained from the one above by taking $m(x, y) = \{z \mid N_{xy}^z \neq 0\}$. This weaker notion of hypergroups does not play any role in this paper, but keeping these two definitions in mind could help the reader looking through the literature.

[10] In a large-$N$ field theory, the fields within an interaction such as (20) are naturally cyclically ordered, since they are effectively inside a trace. In such cases it would be meaningful to consider selection rules dictated by fusion algebras which are not commutative.

**Conjugacy classes as fusion algebras:** Before proceeding, let us pause to note that the construction here generalizes the one discussed above based on the conjugacy classes of a finite group $G$. To see this, let $A = \text{Conj}(G)$ be the set of conjugacy classes of $G$. We now introduce a product structure on $A$ by embedding it into the group algebra $\mathbb{C}[G]$ via

$$\text{Conj}(G) \ni [g] \mapsto \sum_{g' \sim g} g' \in \mathbb{C}[G]. \tag{22}$$

Its images are well-known to form a basis of the center and therefore a subalgebra of $\mathbb{C}[G]$. The conjugation is given by $\overline{[g]} := [g^{-1}]$, with

$$N^{[e]}_{[g],[h]} = \begin{cases} \#[g], & \text{if } [h] = [g^{-1}], \\ 0, & \text{otherwise.} \end{cases} \tag{23}$$

The properties we derive below for general fusion algebras reproduce the results discussed in Sec. 2.1 for the conjugacy classes of $G$, as we will explicitly demonstrate in Sec. 2.3.3 below.

**An artificial field theory for an arbitrary finite hypergroup:** Let us point out that the considerations here allow us to write down a scalar field theory (albeit a rather artificial one) for an arbitrary finite hypergroup $A = \{e, x, y, \ldots\}$ with $n$ elements. To do so, we simply take $n$ scalar fields $\phi_1, \ldots, \phi_n$ as above, and consider a Lagrangian with a cubic interaction term $c^z_{xy} \phi_x \phi_y \overline{\phi_z}$ with nonzero $c^z_{xy}$ for all $x, y, z$ such that $N^z_{xy} \neq 0$. Note that this construction does not require the integrality of $N^z_{xy}$, and therefore applies to hypergroups which are not fusion algebras.

### 2.3.2 Selection rules

**Tree-level properties:** Repeating our previous analysis for conjugacy classes at the level of fusion algebras, it is easy to prove that a tree diagram with fields labeled by elements $x_1, \ldots, x_n$ is a valid scattering process if and only if it satisfies the same condition as for a single vertex. Indeed, we may again proceed by induction. As before, the case of a tree diagram with one vertex is trivial. We next assume that we have proven the statement for $k$ vertices. This means that the external fields $\phi_i$ with $i = 1, \ldots, n+1$ coming into the $k$ vertices are labelled by elements $x_1, x_2, \ldots, x_{n+1}$ of the fusion algebra satisfying

$$e \prec x_1 x_2 \ldots x_{n+1}. \tag{24}$$

We now consider a diagram with $k+1$ vertices. We may split such a diagram into a subdiagram with $k$ vertices and an additional subdiagram with one vertex, connected by the leg $x_{n+1}$, say. We call the external legs of the former $x_i$ as before, and the external legs of the latter $y_1, y_2, \ldots, y_m$. Consistency of the final vertex tells us that

$$e \prec y_1 y_2 \ldots y_m \overline{x_{n+1}}. \tag{25}$$

We now use the property that $x_{n+1}$ is a single element, so that the product of the remaining external legs $y_1, \ldots y_m$ must contain the conjugate of the internal leg $x_{n+1}$, that is

$$x_{n+1} \prec y_1 y_2 \ldots y_m. \tag{26}$$

On the other hand, from (24) we also have

$$\overline{x}_{n+1} \prec x_1 x_2 \ldots x_n. \tag{27}$$

Combining these two equations, we see that

$$e \prec x_{n+1} \overline{x_{n+1}} \prec x_1 x_2 \ldots x_n y_1 y_2 \ldots y_m, \tag{28}$$

and hence the property is proven for $k+1$ vertices. By induction, our statement holds for tree diagrams with any number of vertices.

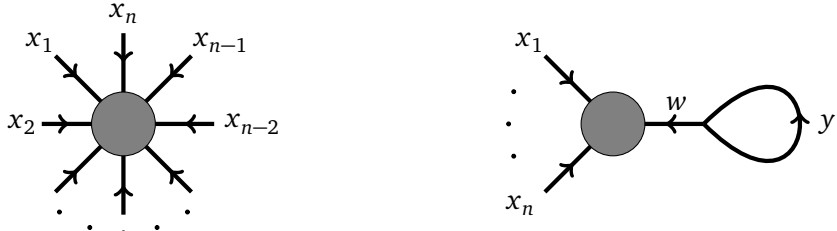

Figure 3: At tree-level (left), scattering amongst $x_1, \ldots, x_n$ is allowed only when the product $x_1 \ldots x_n$ contains the identity $e$, as would be the case for a single vertex. At one-loop (right) on the other hand, the process is allowed as long as the product $x_1 \ldots x_n$ contains an element $w$ that is contained in $y\bar{y}$ for some $y$. As far as our selection rules are concerned, we may always use the associativity of the hypergroup to draw each loop with a single segment and a single outgoing leg, as done above.

**Loop order:** We now consider the fate of our selection rules at non-zero loop order. For a diagram with $L$ loops and with external fields labeled by

$$x_1, \ x_2, \ \ldots, x_n, \tag{29}$$

the first step is again to make $L$ cuts such that we reduce to a connected tree diagram with $2L$ new external fields:

$$y_1, \ \overline{y}_1, \ y_2, \ \overline{y}_2, \ \ldots, y_L, \ \overline{y}_L. \tag{30}$$

Then from the consistency of the tree diagram we find that

$$e \prec x_1 x_2 \ldots x_n (y_1 \overline{y}_1)(y_2 \overline{y}_2) \ldots (y_L \overline{y}_L). \tag{31}$$

This means in particular that we can pick a $\overline{z}_i \prec y_i \overline{y}_i$ such that

$$e \prec x_1 x_2 \ldots x_n \overline{z_1} \cdots \overline{z_L}, \tag{32}$$

which further implies that we have an element $w \prec z_L z_{L-1} \cdots z_1$ such that

$$w \prec x_1 x_2 \ldots x_n. \tag{33}$$

We can then summarize the condition for a non-zero scattering amplitude at $L$ loops in the following manner. Let

$$\text{Com}(A) := \{z \mid z \prec y\overline{y} \text{ for some } y \in A\}, \tag{34}$$

and

$$\text{Com}(A)^L := \{w \mid w \prec z_1 \cdots z_L \text{ for some } z_1, \ldots, z_L \in \text{Com}(A)\}. \tag{35}$$

We then have the following result,

> *The L-loop scattering amplitude with n legs labeled by $x_1, \ldots, x_n$ is non-zero only when*
>
> $$w \prec x_1 \ldots x_n \quad \text{for some } w \in \text{Com}(A)^L. \tag{36}$$

This result can be understood intuitively as follows. First, note that insofar as the selection rules are concerned, we may use the associativity of the hypergroup to rearrange loop diagrams such that each loop is composed of a single segment, and has a single outgoing leg. The result of such a rearrangement is shown for one-loop in the right panel of Figure 3. In this presentation, it is clear that, whereas at tree-level the process is only allowed when the product $x_1 \ldots x_n$ contains the identity $e$, at one-loop level the process is allowed as long as the product $x_1 \ldots x_n$ contains an element $w$ that is contained in $y\bar{y}$ for some $y$. Similar statements apply for higher loop orders.

**The selection rule at large enough loop order:**  As $\text{Com}(A) \subset \text{Com}(A)^2 \subset \cdots$, we see that the conditions for nonzero scattering amplitudes become increasingly more relaxed at higher loop orders. Since $A$ is a finite set, this chain eventually stabilizes to

$$\text{Com}(A)^\infty := \{w \mid w \prec z_1 \cdots z_k \text{ for some } z_1, \ldots, z_k \in \text{Com}(A) \text{ for some } k\}, \tag{37}$$

and the all-loop order condition for a non-zero scattering amplitude is given by

$$w \prec x_1 x_2 \cdots x_n \quad \text{for some } w \in \text{Com}(A)^\infty. \tag{38}$$

We call the smallest $L$ such that $\text{Com}(A)^L = \text{Com}(A)^\infty$ the *conjugate pair length* $\text{cl}(A)$, in analogy with the commutator length of a group.

As we explain in more detail in Appendix A, the condition (38) can be drastically simplified. Let us say that $x \sim y$ for $x, y \in A$ if and only if there exists $w \in \text{Com}(A)^\infty$ such that $x \prec wy$. Roughly, this means that the particle of type $x$ can be transformed into a particle of type $y$ if we allow loop processes at arbitrary loop order. We can show that this relation $\sim$ is an equivalence relation. We then define $\text{Gr}[A] := A/\sim$, and denote its elements by $[x] \in \text{Gr}[A]$. It is known that we can introduce an honest product structure on $\text{Gr}[A]$ via

$$[x][y] = [z], \qquad \text{if and only if } N_{xy}^z \neq 0, \tag{39}$$

which can be shown to make $\text{Gr}[A]$ into a group. We will call $\text{Gr}[A]$ the "groupification" of the hypergroup (or fusion algebra) $A$. As reviewed in Appendix A, $\text{Gr}[A]$ is the universal maximal group for which the relation (39) holds.

In our case, as we start from a commutative $A$, the resulting group $\text{Gr}[A]$ is an Abelian group. Then the condition (38) is equivalent to

$$[e] = [x_1][x_2] \cdots [x_n], \tag{40}$$

i.e. the selection rules at arbitrary loop order reduce to ones coming from the finite Abelian group $\text{Gr}[A]$.

We emphasize that the selection rules obtained here hold for arbitrary theories satisfying the constraints in (20) and (21). We do not expect any accidental restoration of the tree-level selection rules. It could be the case that fields carrying hypergroup labels needed to violate selection rules at loop level are absent in the theory, but we do not consider such "unfaithful" labelings here. Let us also mention that it for any particular theory of the type studied here, it may be possible to derive additional, more stringent selection rules, that are independent of the tree-level selection rules, though we do not describe this here.

### 2.3.3  Comments

**Conjugacy classes as fusion algebras:**  Let us first check that our general formulation using fusion algebras reproduce the results derived in Sec. 2.1. For this, we use $A = \text{Conj}(G)$ introduced in (22) and (23) as the fusion algebra. At tree level, using the fusion algebra, the allowed processes are those such that

$$[e] \prec [g_1][g_2] \cdots [g_n]. \tag{41}$$

Using (22) to embed the fusion algebra inside the group algebra $\mathbb{R}[G]$, the right-hand side is an element

$$\Big(\sum_{g_1' \sim g_1} g_1'\Big)\Big(\sum_{g_2' \sim g_2} g_2'\Big) \cdots \Big(\sum_{g_n' \sim g_n} g_n'\Big). \tag{42}$$

Then the condition (41) is seen to be equivalent to the existence of group elements $g_1' \sim g_1$, $g_2' \sim g_2$, ..., $g_n' \sim g_n$ such that

$$e = g_1' g_2' \cdots g_n', \tag{43}$$

which is exactly the condition (16) that we found in Sec. 2.1.

To see the agreement of the selection rules at the loop order, it suffices to show that $\mathrm{Com}(\mathrm{Conj}(G))$ contains exactly the conjugacy classes of commutators. This follows from the fact that $[g] \in \mathrm{Com}(\mathrm{Conj}(G))$ means that there is an $[h]$ such that $[g] \prec [h][h^{-1}]$. Expanding the definitions, again using (22), we find that this means that

$$g = (k^{-1}hk)(\ell h^{-1}\ell^{-1}), \tag{44}$$

for some $k$ and $\ell$. This in turn means that

$$g = h'\ell k(h')^{-1}k^{-1}\ell^{-1} = [h', \ell k], \tag{45}$$

where $h' := k^{-1}hk$. This process can also be reversed to show that the conjugacy class of any $[g, h]$ is in $\mathrm{Com}(\mathrm{Conj}(G))$.

Mathematically, what we demonstrated here is that the Abelianization of $G$ and the groupification of the fusion algebra $\mathrm{Conj}(G)$ agree, i.e.

$$\mathrm{Gr}[\mathrm{Conj}(G)] = \mathrm{Ab}[G]. \tag{46}$$

**Example for conjugate pair length:**   Beyond the conjugacy class example just discussed, one may ask the following question: for any given fusion algebra, at which loop order does the symmetry get reduced to that at arbitrary order? Since we are unaware of any math literature discussing this topic, we give some concrete examples in Appendices B and C. The former appendix discusses some familiar WZW fusion algebras, while the latter focuses on low-rank fusion algebras with conjugate pair length greater than one.

**Representations as fusion algebras:**   Before proceeding, let us note that irreducible representations $R_i$ of a group $G$ also form a fusion algebra $\mathrm{Rep}(G)$ under the tensor product:

$$R_i \otimes R_j = \bigoplus_k N_{ij}^k R_k. \tag{47}$$

Our analysis is therefore also applicable to field theories with ordinary $G$ symmetry. But it is important to note that we use only very crude information coming from the $G$ symmetry in this formalism—indeed, we use only the decomposition of the tensor product into irreducible summands, and not the Clebsch-Gordan coefficients, which dictate how the individual basis vectors within representations combine.

For example, we can apply the analysis of the selection rules at large enough loop order to the case of $A = \mathrm{Rep}(G)$. This results in an all-loop selection rule based on the Abelian group

$$\mathrm{Gr}[A] = A/\sim = \mathrm{Rep}(Z_G), \tag{48}$$

where $Z_G$ is the center of $G$. This is not wrong, in that we definitely have the symmetry $Z_G \subset G$ at all-loop order. But we also have the even larger $G$ symmetry at all-loop order, meaning that the selection rules at tree level are actually not reduced, even at all-loop order.

### 2.4 String theory realization: Worldsheet non-invertible symmetries

The structure identified in the previous section arises naturally in perturbative string theory. Say, for example, that the internal worldsheet theory has a part described by a rational chiral algebra, whose irreducible representations we denote by $V_1$, $V_2$, .... Here we only need to assume the existence of the chiral algebra on the left-moving side of the worldsheet, say. It is well-known that these irreducible representations satisfy a fusion algebra of the form

$$V_i \hat{\otimes} V_j = \bigoplus_k N_{ij}^k V_k \,, \tag{49}$$

where $\hat{\otimes}$ denotes the fusion product between modules. One-particle states of the theory are then classified by $V_i$, and the tree-level amplitudes with $n$ external particles labeled by $V_{i_1}$, ..., $V_{i_n}$ are nonzero only when $V_{i_1} \hat{\otimes} \cdots \hat{\otimes} V_{i_n}$ contains the identity representation. This is indeed the structure we have found above.

In the previous paragraph we assumed the existence of a rational chiral algebra on the worldsheet, but this is not actually necessary. All we need is a non-invertible symmetry in the worldsheet theory $T$, described by a fusion category $C$.[11] Then the states of the worldsheet theory $T$ on $S^1$ can be decomposed in terms of the Drinfeld double $Z(C)$ of $C$, see e.g. [16]. The simple objects of $Z(C)$ describe not only states in the untwisted sector, but also states in the sector twisted by various simple objects of $C$. As perturbative string theory with a fixed worldsheet theory $T$ in the usual sense only uses the untwisted sector of $T$, we do not have to use all of the objects of $Z(C)$; we only need to use the simple objects of $Z(C)$ which appear in the decomposition of the untwisted sector.

For example, let us say that the worldsheet theory $T$ has a fusion category symmetry $C$ which is actually a finite group $G$. For ease of description, let us assume that its worldsheet anomaly is zero. Then $Z(C)$ is the Drinfeld double of $G$, and the only part which appears in the decomposition of the untwisted sector is the subcategory $\text{Rep}(G)$, which controls the selection rules of the scattering amplitudes. This is just an extremely pretentious way of saying that when the worldsheet theory has a finite group symmetry $G$, the one-particle states of the spacetime theory carry an action of $G$ and therefore are decomposed into representations of $G$.

Now let us instead take $T' = T/G$, the non-Abelian orbifold by $G$, as the worldsheet theory. This theory $T'$ has $\text{Rep}(G)$ as the fusion category describing its non-invertible symmetry [17]. The Drinfeld double is $Z(\text{Rep}(G)) = Z(G)$, but now the simple objects appearing in the untwisted sector of $T' = T/G$ are actually the $G$-twisted sectors of $T$, and are labeled by $\text{Conj}(G)$. This indeed reproduces the fact we saw in Sec. 2.2 that the scattering amplitudes of a non-Abelian orbifold theory are controlled by conjugacy classes of $G$.

## 3 Case studies

In the previous section we described the general structure of our selection rules in an abstract manner. In the current section we explore various explicit examples illustrating our results.

### 3.1 ADE orbifolds

We begin our study of concrete selection rules by analyzing some familiar non-Abelian orbifolds in string theory, namely those of the form $\mathbb{C}^2/\Gamma$ where $\Gamma$ is a finite subgroup of $SU(2)$. Such $\Gamma$'s are well-known to have an ADE classification, and are double covers of finite subgroups of

---

[11]Other results on the topic of worldsheet non-invertible symmetries can be found in [8, 15].

$SO(3)$, known as binary polyhedral groups. For more background information, see e.g. [18, Appendix A]. As the A cases are Abelian, we will concentrate on the D and E cases here.

**The $D_{n+2}$ case: binary dihedral groups**  In this case the group has $4n$ elements, and is generated by $a$, $b$, $c$, satisfying $a^n = b^2 = c^2 =: z$, $c = ab = ba^{-1}$, $z^2 = e$. This is the double cover of the dihedral group $D_{2n}$ of $2n$ elements, obtained by setting $z = e$. Somewhat confusingly, this corresponds to $D_{n+2}$ in the ADE classification. There are $n + 3$ conjugacy classes, $[e]$, $[z]$, $[a]$, ..., $[a^{n-1}]$, $[b]$, and $[c]$. For the case of $n = 2$, in which the group becomes the quaternion group $Q_8$, the product table is given by

$$
\begin{pmatrix}
e & z & a & b & c \\
z & e & a & b & c \\
I & a & 2e + 2z & 2c & 2b \\
b & b & 2c & 2e + 2z & 2a \\
c & c & 2b & 2a & 2e + 2z
\end{pmatrix},
\tag{50}
$$

where by abuse of notation we denote each conjugacy classes by a representative element. The case of more general $n$ can be worked out analogously.

We denote the image of each generator in the abelianization by the corresponding upper-case letter. Then the abelianization is given by $Z = E$, $A^2 = E$, $C = AB$, and $B^2 = E$ or $B^2 = A$ depending on whether $n$ is even or odd. Therefore the result is $\mathbb{Z}_2 \times \mathbb{Z}_2$ when $n$ is even and $\mathbb{Z}_4$ when $n$ is odd. The commutator length is 1, and therefore starting at one-loop order the preserved symmetry is either $\mathbb{Z}_2 \times \mathbb{Z}_2$ or $\mathbb{Z}_4$.

**The $E_6$ case: binary tetrahedral group**  In this case the group has 24 elements, and is generated by $a$, $b$, $c = ab$, satisfying $a^3 = b^3 = c^2 =: z$, $z^2 = e$. The elements $a$, $b$, $c$ correspond to rotations around the center of a face, around a vertex, and around the center of an edge, of a tetrahedron. The seven conjugacy classes are $[e]$, $[z]$, $[a]$, $[a^2]$, $[b]$, $[b^2]$, and $[c]$. The product table is as follows:

$$
\begin{pmatrix}
e & z & a & a^2 & b & b^2 & c \\
z & e & b^2 & b & a^2 & a & c \\
a & b^2 & a^2+3b & 4z+2c & 4e+2c & 3a^2+b & 3a+3b^2 \\
a^2 & b & 4z+2c & 3a+b^2 & a+3b^2 & 4e+2c & 3a^2+3b \\
b & a^2 & 4e+2c & a+3b^2 & 3a+b^2 & 4z+2c & 3a^2+3b \\
b^2 & a & 3a^2+b & 4e+2c & 4z+2c & a^2+3b & 3a+3b^2 \\
c & c & 3a+3b^2 & 3a^2+3b & 3a^2+3b & 3a+3b^2 & 6e+6z+4c
\end{pmatrix}.
\tag{51}
$$

The commutator subgroup is of order 8, and is isomorphic to the quaternion group. The abelianization is $\mathbb{Z}_3$, where $C = Z = E$ and $B = A^{-1}$. The commutator length is 1, and therefore starting at one-loop order the preserved symmetry is $\mathbb{Z}_3$.

We can explicitly see that the multiplication rule in (51) respects the $\mathbb{Z}_3$ symmetry which assigns charge $+1$ to $a$, charge $-1$ to $b$, and charge $0$ to $e$, $z$, $c$. At the same time, we see that this multiplication table contains more detailed information about the tree-level scattering processes.

**The $E_7$ case: binary octahedral group**  In this case the group has 48 elements, and is generated by $a$, $b$, $c = ab$, satisfying $a^4 = b^3 = c^2 =: z$, $z^2 = e$. The elements $a$, $b$, $c$ correspond to rotations around the center of a face, around a vertex, and around the center of an edge, of an octahedron. The eight conjugacy classes are $[e]$, $[z]$, $[a]$, $[a^2]$, $[a^3]$, $[b]$, $[b^2]$, and $[c]$, where the multiplication table is omitted as it is not too enlightening. The commutator subgroup is

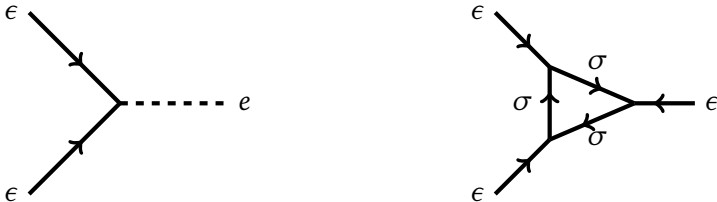

Figure 4: At tree-level, the fusion algebra requires that diagrams with only external $e$- and $\epsilon$-sector particles must have an even number of the latter. On the other hand, at one loop configurations with an odd number are allowed, due to the existence of the $\epsilon\sigma\sigma$ vertex.

of order 24, and is isomorphic to the binary tetrahedral group. The abelianization is $\mathbb{Z}_2$, where $B = Z = E$, $C = A$, and $A^2 = E$. The commutator length is 1, and therefore starting at one-loop order the preserved symmetry is $\mathbb{Z}_2$.

**The $E_8$ case: binary icosahedral group**   In this case the group has 120 elements, and is generated by $a$, $b$, $c = ab$, and $z = c^2$ satisfying $a^5 = b^3 = c^2 =: z$, $z^2 = e$. The elements $a$, $b$, $c$ correspond to rotations around the center of a face, around a vertex, and around the center of an edge, of an icosahedron. The nine conjugacy classes are $[e]$, $[z]$, $[a]$, $[a^2]$, $[a^3]$, $[b]$, $[b^2]$, and $[c]$. The commutator subgroup is equal to the original group, and the abelianization is trivial. The commutator length is 1, and therefore starting at one-loop order there is no preserved symmetry.

**Summary**   In all cases the commutator length is 1, and hence the selection rule stabilizes already at one loop. Furthermore, the abelianization, which is always a rather small Abelian group, matches with the center of the corresponding ADE group, which is a manifestation of the McKay correspondence.[12] Nevertheless, there are many conjugacy classes which form a complicated fusion algebra, leading to nontrivial selection rules at tree level.

## 3.2   The Ising theory

Next, let us consider an example of selection rules coming from a fusion algebra that does not come from a group. Let us suppose that the internal worldsheet theory has a symmetry given by the Ising fusion category. For simplicity of presentation, we assume that the internal worldsheet theory is actually a direct product of the $c = 1/2$ Ising model and the rest, but the analysis below is applicable to more general cases as well, such as $Spin(7)$ compactifications of the heterotic string [21].

Recall that the Ising theory has Virasoro primaries 1, $\sigma$, and $\epsilon$, whose dimensions are given by $(h, \bar{h}) = (0, 0)$, $(1/16, 1/16)$, and $(1/2, 1/2)$, respectively. Correspondingly, the spacetime fields can be decomposed into three sectors $e$, $\sigma$, and $\epsilon$, whose fusion algebra $A$ has the fusion rules $\epsilon^2 = e$, $\epsilon\sigma = \sigma$, and $\sigma^2 = e + \epsilon$. From this, we easily conclude that at tree level, interaction terms of the form $O_n := \epsilon^n$ are allowed only when $n$ is even.

---

[12]If we consider the compactification of IIB on $\mathbb{C}^2/\Gamma$, then the resulting 6D (2,0) SCFT has a 2-form *defect group* that is also given by Ab[$\Gamma$] [19], whose charged objects are non-dynamical surface defects coming from D3-branes wrapping relative cycles in the resolved $\mathbb{C}^2/\Gamma$. In that context, the twisted sectors for strings on $\mathbb{C}^2/\Gamma$ were shown in [20] to correspond to exceptional divisors of $\mathbb{C}^2/\Gamma$, and the fact that their scattering admits an Ab[$\Gamma$] charge conservation rule is required for the entire scattering process (at any loop order) to admit a consistent charge pairing with the non-dynamical surface defects.

What happens at loop order? Applying the analysis of Sec. 2.3, we see that $\text{Com}(A) = \text{Com}(A)^{\infty} = \{e, \epsilon\}$, and therefore the all-loop selection rules are already realized at one loop, and are those for the group $\text{Gr}[A] = \{[e], [\sigma]\}$ with $[\sigma]^2 = [e]$, i.e. those for a $\mathbb{Z}_2$ symmetry.

Intuitively, the breakdown of the selection rules at loop level may be understood as in Figure 4. Consider a tree-level diagram with external particles in the $e$ or $\epsilon$ sectors. For any such diagram, the fusion algebra gives a selection rule requiring that there be an even number of external $\epsilon$ particles. But at one loop, the allowed $\epsilon\sigma\sigma$ vertex may be used to write diagrams violating this selection rule. On the other hand, the fusion algebra does not allow us to draw diagrams with an odd number of external $\sigma$ particles at any loop order, which corresponds to the all-loop $\text{Gr}[A] = \mathbb{Z}_2$ selection rule.

Thus far, we have obtained our selection rules by assigning elements of the fusion algebra to spacetime sectors. An alternative way to obtain selection rules is to study the action of the Ising category on the worldsheet fields. Recall that each primary of the Ising model has a corresponding Verlinde line operator, which we denote by $L_e$, $L_\sigma$, and $L_\epsilon$, whose fusion algebra is given by $L_\epsilon^2 = L_e$, $L_\epsilon L_\sigma = L_\sigma$, and $L_\sigma^2 = L_e + L_\epsilon$. The action of these line operators on the primaries is given in the following table:

$$
\begin{array}{c|ccc}
 & e & \sigma & \epsilon \\
\hline
L_e & +1 & +1 & +1 \\
L_\sigma & \sqrt{2} & 0 & -\sqrt{2} \\
L_\epsilon & +1 & -1 & +1
\end{array}
\,,
\tag{52}
$$

as readily follows from the $S$ matrix of this modular tensor category. In this point of view, the all-loop $\mathbb{Z}_2$ symmetry we found before can be understood as being the selection rule coming from the $\mathbb{Z}_2$ symmetry generated by $L_\epsilon$.

We should emphasize the distinction between the two approaches above. In the first approach, we label the spacetime sectors by representations of the categorical symmetry $C$, i.e. by elements of a fusion algebra $A \subset Z(C)$, which gives constraints on the spacetime action in the way discussed in Sec. 2. In the second approach, we use the action of $C$ on the worldsheet fields to obtain constraints on scattering amplitudes, which are in turn translated to constraints on the spacetime action. In general, the constraints obtained via these two approaches are not completely equivalent, as already follows from the discussion of ordinary $G$ symmetry at the end of Sec. 2.3. In the Ising case studied above, the all-loop results following from the first approach were seen to be equivalent to the selection rules coming from the invertible subgroup of $C$ (generated by $L_e$) in the second approach.

A similar equality holds whenever the worldsheet theory has a modular tensor category $\mathcal{M}$ as the symmetry. Indeed, the first approach leads to spacetime sectors controlled by the fusion algebra $M$ of $\mathcal{M}$, which gives rise to tree-level selection rules as in Sec. 2.3. The all-loop selection rule is then given by the groupification $M \to \text{Gr}[M] = M/\sim$. On the other hand, in the second approach we consider the group $\text{Inv}(\mathcal{M})$ formed by invertible line operators of $\mathcal{M}$. As $\mathcal{M}$ is braided, $\text{Inv}(\mathcal{M})$ is an Abelian group, and the spacetime sectors can be labeled by its Pontryagin dual $\widehat{\text{Inv}(\mathcal{M})}$. It is known [22, Corollary 8.22.8] that

$$
\widehat{\text{Inv}(\mathcal{M})} = \text{Gr}[M],
\tag{53}
$$

i.e. the all-loop selection rule arising from the fusion algebra of the sectors is simply the selection rule coming from the invertible subgroup of the full non-invertible symmetry.

### 3.3 Selection rules for $S^1/\mathbb{Z}_2$

We now discuss bosonic string theory on $S^1_R/\mathbb{Z}_2$, where the subscript $R$ specifies the radius of the circle, and the $\mathbb{Z}_2$ acts by flipping the coordinate. This turns out to be a nice illustration of both the selection rules for non-Abelian orbifolds, as well as those for the Ising category.

#### 3.3.1 As Abelian orbifolds

We first discuss the standard presentation of $S^1_R/\mathbb{Z}_2$ as an Abelian orbifold by $\mathbb{Z}_2 = \{e, g\}$. The closed string sector is decomposed into the untwisted sector (corresponding to $[e]$) and the twisted sector (corresponding to $[g]$). The interactions preserve the $\mathbb{Z}_2$ symmetry under which $[e]$ is even and $[g]$ is odd.

The description above is very crude, in that the strings stuck at different fixed points of $S^1/\mathbb{Z}_2$ are both assigned to the class $[g]$. These can be distinguished in our framework using the equality

$$S^1_R/\mathbb{Z}_2 = S^1_{2R}/(\mathbb{Z}_2 \times \mathbb{Z}_2), \tag{54}$$

where the first $\mathbb{Z}_2 = \{e, s\}$ acts on $S^1_{2R}$ by a half-shift and the second $\mathbb{Z}_2 = \{e, g\}$ acts by flipping the coordinate. This is still an Abelian orbifold, and there are four sectors labeled by $[e]$, $[s]$, $[g]$, and $[gs]$. The interactions preserve the $\mathbb{Z}_2 \times \mathbb{Z}_2$ symmetry acting on these sectors. In the usual terminology, the states in the $[e]$- and $[s]$-twisted sectors are the untwisted strings with even and odd winding numbers, while the states in the $[g]$- and $[gs]$-twisted sectors are the twisted sectors localized at the two fixed points.

This implies, for example, that the merger of two twisted sector states at the same fixed point produces untwisted sector states of even winding number, i.e. $[g][g] = [gs][gs] = [e]$. Similarly, the merger of two twisted sector states at two different fixed points produces untwisted sector states of odd winding number, i.e. $[g][gs] = [s]$.

#### 3.3.2 As non-Abelian orbifolds

By generalizing the trick used above, it is possible to regard $S^1_R/\mathbb{Z}_2$ as a non-Abelian orbifold and hence to derive more detailed selection rules, albeit this time only applicable at tree level. Indeed, we have

$$S^1_R/\mathbb{Z}_2 = S^1_{2nR}/D_{4n}, \tag{55}$$

where $D_{4n} := \mathbb{Z}_{2n} \rtimes \mathbb{Z}_2$ is the dihedral group of order $4n$. We use the notation $\mathbb{Z}_2 = \{e, g\}$ and $\mathbb{Z}_{2n} = \{e, s, \ldots, s^{2n-1}\}$, such that $gsg = s^{-1}$. The conjugacy classes are

$$[e], \ [s] = \{s, s^{-1}\}, \ \ldots, \ [s^{n-1}] = \{s^{n-1}, s^{1-n}\}, \ [s^n], \tag{56}$$

and

$$[g] = \{g, gs^2, \cdots, gs^{2n-2}\}, \quad [gs] = \{gs, gs^3, \cdots, gs^{2n-1}\}. \tag{57}$$

In the usual terminology, the $[s^k]$-twisted sector states are those whose winding numbers are $\pm k \bmod 2n$, and the $[g]$- and $[gs]$-twisted sector states are those which are stuck at each of the two fixed points on $S^1/\mathbb{Z}_2$. The fusion rules such as

$$[s][s^2] = [s] + [s^3], \tag{58}$$

then constrain the scattering processes. But since the group is now non-Abelian, these selection rules are applicable only at tree level. As the commutator length of dihedral groups is one, the selection rules reduce to the all-loop ones already at one loop. Since the abelianization map gives

$$D_{4n} = \mathbb{Z}_{2n} \rtimes \mathbb{Z}_2 \to \mathbb{Z}_2 \times \mathbb{Z}_2, \tag{59}$$

the resulting selection rule is the one we already studied using (54) above.

### 3.3.3 The $D_8$ symmetry

Let us now study the symmetries of the worldsheet theory for $S^1/\mathbb{Z}_2$ in more detail. We denote by $\Phi_{m,w}$ the momentum $m$, winding $w$ local operator in the theory before orbifolding, where $m, w \in \mathbb{Z}$. Then the linear combinations

$$\widehat{\Phi}_{m,w} := \frac{1}{\sqrt{2}}\left(\Phi_{m,w} + \Phi_{-m,-w}\right), \tag{60}$$

survive the orbifold; their scaling dimensions are given by

$$(h, \bar{h}) = \left(\frac{1}{2}\left(\frac{m}{R} + \frac{wR}{2}\right)^2, \frac{1}{2}\left(\frac{m}{R} - \frac{wR}{2}\right)^2\right). \tag{61}$$

Denote by $\tau_1$ and $\tau_2$ the worldsheet twist fields of dimension $(\frac{1}{16}, \frac{1}{16})$ associated to the two fixed points. These twist fields satisfy OPEs schematically of the form

$$\tau_1 \cdot \tau_1 \sim \sum_{i,j} C_{2i,2j}\widehat{\Phi}_{2i,2j} + \sum_{i,j} C_{2i+1,2j}\widehat{\Phi}_{2i+1,2j}, \tag{62}$$

$$\tau_2 \cdot \tau_2 \sim \sum_{i,j} C_{2i,2j}\widehat{\Phi}_{2i,2j} - \sum_{i,j} C_{2i+1,2j}\widehat{\Phi}_{2i+1,2j}, \tag{63}$$

$$\tau_1 \cdot \tau_2 \sim \sum_{i,j} C_{2i,2j+1}\widehat{\Phi}_{2i,2j+1}, \tag{64}$$

as was detailed e.g. in [23].

In the notation of Sec. 3.3.1, i.e. when we view this system as the orbifold $S^1_{2R}/(\mathbb{Z}_2 \times \mathbb{Z}_2)$, the fields $\widehat{\Phi}_{i,2j}$ are in the sector $[e]$ and the fields $\widehat{\Phi}_{i,2j+1}$ are in the sector $[s]$, while $\tau_1$ is in the sector $[g]$ and $\tau_2$ is in the sector $[gs]$. The OPE given above satisfies the selection rules discussed there. In [23], it was noticed that the worldsheet actually has a $D_8$ symmetry[13] generated by two order-2 operations

$$b: \quad (\tau_1, \tau_2, \widehat{\Phi}_{m,w}) \mapsto (-\tau_1, \tau_2, (-1)^w\widehat{\Phi}_{m,w}), \tag{65}$$

$$c: \quad (\tau_1, \tau_2, \widehat{\Phi}_{m,w}) \mapsto (\tau_2, \tau_1, (-1)^m\widehat{\Phi}_{m,w}). \tag{66}$$

Note that the operation

$$a = bc: \quad (\tau_1, \tau_2, \widehat{\Phi}_{m,w}) \mapsto (-\tau_2, \tau_1, (-1)^{m+w}\widehat{\Phi}_{m,w}), \tag{67}$$

is of order 4. Furthermore, the operation

$$a^2: \quad (\tau_1, \tau_2, \widehat{\Phi}_{m,w}) \mapsto (-\tau_1, -\tau_2, \widehat{\Phi}_{m,w}), \tag{68}$$

together with $b$ forms a $\mathbb{Z}_2 \times \mathbb{Z}_2 \subset D_8$ subgroup distinguishing the four sectors $[e]$, $[s]$, $[g]$, and $[gs]$ above. We emphasize that this $D_8$ symmetry is distinct from the $D_8$ used when we regarded the system as $S^1_{4R}/D_8$ in Sec. 3.3.2.

### 3.3.4 Non-invertible symmetries at generic radius

The $S^1/\mathbb{Z}_2$ model has also a continuum of non-invertible symmetries, which we review following [26–28]. For every $U(1)_m \times U(1)_w$ rotation $U_{(\theta,\phi)}$ in the original $S^1$ theory, we obtain a non-invertible symmetry in the $S^1/\mathbb{Z}_2$ gauge theory, generated by

$$\hat{U}_{(\theta,\phi)} = U_{(\theta,\phi)} + U_{(-\theta,-\phi)}, \tag{69}$$

---

[13]This worldsheet symmetry has been studied from a spacetime perspective in [24, 25]. More recent accounts can be found in [26–28].

all of which have quantum dimension 2. The fusion rules are

$$\hat{U}_{(\theta,\phi)}\hat{U}_{(\theta',\phi')} = \hat{U}_{(\theta+\theta',\phi+\phi')} + \hat{U}_{(\theta-\theta',\phi-\phi')}. \tag{70}$$

These symmetries act on the operators $\widehat{\Phi}_{m,w}$ as

$$\hat{U}_{(\theta,\phi)}: \quad \widehat{\Phi}_{m,w} \to 2\cos(m\theta + w\phi)\widehat{\Phi}_{m,w}. \tag{71}$$

When $\theta, \phi \in \{0, \pi\}$, the operators $U_{(\theta,\phi)}$ are themselves invariant under the $\mathbb{Z}_2$ used in the orbifolding. In such cases, we do not have to form a sum as in (69). Instead, to fully specify the line operator in the orbifolded theory we must pick a charge $\pm 1$ under this $\mathbb{Z}_2$, giving eight operators

$$U^{\pm}_{(0,0)}, \qquad U^{\pm}_{(0,\pi)}, \qquad U^{\pm}_{(\pi,0)}, \qquad U^{\pm}_{(\pi,\pi)}. \tag{72}$$

Here, $U^{-}_{(0,0)}$ is the generator acting by $-1$ on the twisted sector fields and can be identified with $a^2$ given in (68), i.e.

$$U^{+}_{(0,0)} = e, \qquad U^{-}_{(0,0)} = a^2. \tag{73}$$

We then have

$$U^{\mp}_{(\theta,\phi)} = a^2 U^{\pm}_{(\theta,\phi)}, \tag{74}$$

in general for $\theta, \phi \in \{0, \pi\}$.

Other elements in (72) can similarly be identified with the $D_8$ elements introduced in (65), (66), and (67). As $U_{(\theta,\phi)}$ for $\theta, \phi \in \{0, \pi\}$ acts on the untwisted sector fields as

$$\widehat{\Phi}_{m,w} \mapsto e^{im\theta + iw\theta}\widehat{\Phi}_{m,w}, \tag{75}$$

we see that they correspond to, for example,

$$\{U^{+}_{(\pi,0)}, U^{-}_{(\pi,0)}\} = \{ab, a^3b\}, \quad \{U^{+}_{(0,\pi)}, U^{-}_{(0,\pi)}\} = \{b, a^2b\}, \quad \{U^{+}_{(\pi,\pi)}, U^{-}_{(\pi,\pi)}\} = \{a, a^3\}, \tag{76}$$

where the assignments within each pair are not unique due to the nontriviality of the extension by $\mathbb{Z}_2$. We also note that the equality (69) when $\theta, \phi \in \{0, \pi\}$ should be interpreted as follows:[14]

$$\hat{U}_{(\theta,\phi)} = U^{+}_{(\theta,\phi)} + U^{-}_{(\theta,\phi)}. \tag{77}$$

We now note that the operators

$$U^{\pm}_{(0,0)}, \qquad U^{\pm}_{(0,\pi)}, \qquad \text{and} \qquad \hat{U}_{(0,k\pi/n)} \text{ for } k = 1, \ldots, n-1, \tag{78}$$

have the same fusion rules as $\text{Rep}(D_{4n})$. In fact they generate precisely the $\text{Rep}(D_{4n})$ symmetry arising when we regard $S^1_R/\mathbb{Z}_2$ as $S^1_{2nR}/D_{4n}$ as in Sec. 3.3.2. To see this, we first regard $S^1_R = S^1_{2nR}/\mathbb{Z}_{2n}$. The $\text{Rep}(\mathbb{Z}_{2n})$ symmetry of the left-hand side should measure the winding number of $S^1_R$ modulo $2n$, which means that it is given by $U_{(0,2\pi k/2n)}$, for $k = 0, \ldots, 2n-1$. We then form $S^1_R/\mathbb{Z}_2 = (S^1_{2nR}/\mathbb{Z}_{2n})/\mathbb{Z}_2$. This affects the symmetry generators as we described above, so the simple objects of the natural $\text{Rep}(D_{4n})$ symmetry are indeed the ones listed in (78).

Let us take $n = 2$ here and introduce the new notation

$$\mathcal{N} := \hat{U}_{0,\pi/2}. \tag{79}$$

---

[14]This is because the $\mathbb{Z}_2$ orbifold exchanges the two summands of (69) as in $\begin{pmatrix} 0 & 1 \\ 1 & 0 \end{pmatrix}$, which can be diagonalized to give $\begin{pmatrix} +1 & 0 \\ 0 & -1 \end{pmatrix}$ instead, when $\theta, \phi \in \{0, \pi\}$.

This $\mathcal{N}$ extends the $\mathbb{Z}_2 \times \mathbb{Z}_2$ generated by $a^2$ and $b$, with the fusion rules

$$\mathcal{N}^2 = 1 + a^2 + b + a^2 b, \tag{80}$$

and

$$a^2 \mathcal{N} = b\mathcal{N} = \mathcal{N}. \tag{81}$$

In contrast, the $D_8$ symmetry we discussed above in Sec. 3.3.3 extends the same $\mathbb{Z}_2 \times \mathbb{Z}_2$ symmetry by $a$. This shows that $a$, $b$, and $\mathcal{N}$ together generate an interesting mixture of $D_8$ and $\text{Rep}(D_8)$. By studying the action on $\widehat{\Phi}_{m,w}$, we see that $c = ba$ and $\mathcal{N}$ satisfy $c\mathcal{N} = \mathcal{N}c$.

### 3.3.5 Ising symmetry at a specific radius

At certain special rational points, the worldsheet non-invertible symmetry can be enhanced even further. As an example, we consider the case of $R = 1$.[15] At this radius, the operators $\widehat{\Phi}_{0,2}$ and $\widehat{\Phi}_{1,0}$ have scaling dimensions $(h, \bar{h}) = (1/2, 1/2)$. In fact, at this radius, the theory is known to be equivalent to two copies of the $c = 1/2$ Ising theory.

The mapping between the operators of $S^1_R/\mathbb{Z}_2$ and those of two copies of the Ising theory was given e.g. in [23]. We already saw in Sec. 3.2 that the Ising theory has sectors $e$, $\sigma$, and $\epsilon$, with corresponding Verlinde line operators $L_e$, $L_\sigma$, and $L_\epsilon$. We continue to use the same symbols here, with an additional subscript $1, 2$ to distinguish the two copies. The end result is that the two Ising primaries $\sigma_{1,2}$ of dimension $(1/16, 1/16)$ can be identified with our twisted sector fields $\tau_{1,2}$, and that the two Ising primaries $\epsilon_{1,2}$ of dimension $(1/2, 1/2)$ can be identified as follows,

$$\epsilon_1 := \widehat{\Phi}_{0,2} + \widehat{\Phi}_{1,0}, \qquad \epsilon_2 := \widehat{\Phi}_{0,2} - \widehat{\Phi}_{1,0}. \tag{82}$$

Comparing our $D_8$ transformation rules (65) – (67), we find that the $\mathbb{Z}_2 \times \mathbb{Z}_2$ subgroup generated by $a^2$ and $b$ are related to the two $\mathbb{Z}_2$ symmetries of the two copies of the Ising theory via

$$(L_\epsilon)_1 = b, \qquad (L_\epsilon)_2 = a^2 b, \tag{83}$$

and the exchange of the two Ising factors is the operation $c$ of $D_8$.

We can also see that the non-invertible line $\mathcal{N}$ introduced in (79) has the identification

$$(L_\sigma)_1 (L_\sigma)_2 = c\mathcal{N}. \tag{84}$$

We see that the fusion rule (80) is reproduced from $(L_\sigma)^2 = 1 + L_\epsilon$ and (83).

As we saw in Sec. 3.2, the Ising fusion algebra leads to the selection rule that the interaction $\epsilon^n$ is allowed at tree level only when $n$ is even. This results in the following somewhat interesting spacetime selection rule. First, note that the Virasoro primary $\epsilon_1 \epsilon_2$ has dimension $(1, 1)$ and captures the change in the radius of the $S^1$. In spacetime, it corresponds to a massless radion field. Then, at tree level, the interaction term $(\epsilon_1 \epsilon_2)^n$ is allowed only when $n$ is even. On the other hand, at one loop this selection rule can be violated. If we used only the invertible symmetry, this conclusion would be hard to come by, since $\epsilon_1 \epsilon_2$ is invariant under the entire $D_8$ symmetry—the generator $b$ leaves both $\epsilon_1$ and $\epsilon_2$ invariant, while the generator $a$ exchanges $\epsilon_1$ with $\epsilon_2$, so that their product $\epsilon_1 \epsilon_2$ remains invariant.[16]

---

[15]We can equally consider the T-dual radius $R = 2$, but $R = 1$ matches our previous discussions better.

[16]In fact, for the specific operator $O := \partial X \bar{\partial} X$ of dimension $(1, 1)$, the tree-level interaction term $O^n$ vanishes for odd $n$ for arbitrary values of the radius $R$. This is due to the fact that the subalgebra generated by $O$ within the full operator algebra of the theory is common to all $R$, and therefore the selection rule derived at a given value of $R$ is applicable at any other value of $R$. This argument does not apply to generic operators in the sector $\epsilon_1 \epsilon_2$ of two copies of the Ising fusion algebra, and we expect that the interaction would be nonzero for odd $n$ in the generic case.

## 3.4 Tambara-Yamagami of $\mathbb{Z}_3$

Thus far, all of our explicit examples have involved worldsheet non-invertible symmetries forming a modular tensor category. Such cases are particularly simple, since the spacetime sectors have fusion rules identical to the symmetries themselves. On the other hand, when the worldsheet symmetries do *not* form a modular tensor category, we must instead label the spacetime sectors by representations of the non-invertible symmetry, i.e. by elements of the Drinfeld double of the original fusion category.

In this section we analyze arguably the simplest example of a non-modular non-invertible symmetry, namely the Tambara-Yamagami category $\mathrm{TY}(\mathbb{Z}_3)$. This fusion category has rank-4 with fusion rules given by

$$\eta^3 = e, \qquad \eta \times \mathcal{N} = \mathcal{N}, \qquad \mathcal{N} \times \mathcal{N} = e + \eta + \eta^2. \tag{85}$$

It arises as a symmetry of the 3-states Potts model and the $\mathbb{Z}_3$ parafermion theory; in the context of the string worldsheet, it is relevant for certain Gepner models $(p_1, \ldots p_n)$ with $p_i$ being 1.

The Drinfeld double for $\mathrm{TY}(\mathbb{Z}_N)$ for generic $N$ was studied in the mathematics literature in [29,30]; a more physically motivated discussion can be found in [31]. For the case of $N = 3$, the result is as follows. First, $Z(\mathrm{TY}(\mathbb{Z}_3))$ has a total of 15 objects:

- 6 invertible objects $X_{g,i}$ with $g \in \mathbb{Z}_3$ and $i \in \mathbb{Z}_2$,

- 3 objects of quantum dimension 2 denoted by $Y_{[0,1]}, Y_{[0,2]}, Y_{[1,2]}$ (symmetric in the subscripts),

- 6 objects of quantum dimension $\sqrt{3}$ denoted by $Z_{g,i}$ with $g \in \mathbb{Z}_3$ and $i \in \mathbb{Z}_2$.

The full set of fusion rules amongst these objects can be found in [31], and are as follows,

$$\begin{aligned}
X_{g,i} \times X_{h,j} &= X_{g+h,i+j}, \\
X_{g,i} \times Y_{[g',h']} &= Y_{[g+g',g+h']}, \\
X_{g,i} \times Z_{h,j} &= Z_{2g+h,i+j+\delta_{h,0}+\delta_{2g+h,0}}, \\
Y_{[g,h]} \times Z_{g',i} &= Z_{g+h+g',0} + Z_{g+h+g',1}, \\
Y_{[g,h]} \times Y_{[g',h']} &= \begin{cases} X_{g+g',0} + X_{g+g',1} + Y_{[h+g',g+h']}, & g+g' = h+h', \\ X_{h+g',0} + X_{h+g',1} + Y_{[g+g',h+h']}, & \text{otherwise}, \end{cases} \\
Z_{g,i} \times Z_{h,j} &= X_{-g-h,i+j+\delta_{g,0}+\delta_{h,0}} + Y_{[-g-h+1,-g-h-1]}.
\end{aligned} \tag{86}$$

We now label the spacetime sectors by the elements above. In particular, untwisted sectors must be closed under fusion, and thus must be labelled by a closed subalgebra. An example of such a subalgebra is the set of invertible lines. Besides that, the largest proper subalgebra contains five objects $X_{0,0}, X_{0,1}, Y_{[1,2]}, Z_{0,0}$, and $Z_{0,1}$. Using the abbreviated notation

$$X_{0,0} \to e, \qquad X_{0,1} \to X, \qquad Y_{[1,2]} \to Y, \qquad Z_{0,0} \to Z, \qquad Z_{0,1} \to XZ, \tag{87}$$

the fusion rules amongst them are found to be,

$$X^2 = e, \qquad X \times Y = Y, \qquad Y^2 = e + X + Y, \qquad Y \times Z = Z + XZ, \qquad Z^2 = e + Y. \tag{88}$$

These are none other than the fusion rules of $SU(2)_4$. In the presence of a $\mathrm{TY}(\mathbb{Z}_3)$ symmetry on the worldsheet, we may then label the spacetime sectors by elements of a non-trivial subring of the $SU(2)_4$ fusion ring, of which there are three: $\mathbb{Z}_2 = \langle X \rangle$, $\mathrm{Rep}(D_3) = \langle X, Y \rangle$, and $SU(2)_4$

itself. In all cases the tree-level fusions are dictated by those given above. Assuming that it is the maximum subalgebra that is realized, we have

$$\text{Com}(A) = \text{Com}(A)^\infty = \{e, X, Y\}, \tag{89}$$

from which we see that these selection rules would be broken at one loop and beyond to those for the group $\text{Gr}[A] = \{[e], [Z]\} = \mathbb{Z}_2$. This would mean that the equality in (53) is no longer satisfied.

## Acknowledgments

YT thanks discussions with Satoshi Shirai, and HYZ thanks discussions with Liang Kong and Adar Sharon. JK thanks Jan Albert and Julio Parra-Martinez for useful conversations.

**Funding information**  YT and HYZ are supported in part by WPI Initiative, MEXT, Japan at Kavli IPMU, the University of Tokyo. JK is supported in part by the Department of Energy.

## A  Basics of finite hypergroups

Here we summarize the bare minimum on the theory of finite hypergroups, since it is not easy to find a self-contained reference where only finite ones are discussed. The discussions here are based on [32], [33], [22, Sec. 3], and [34, Sec. 2]; the authors claim no originality in presentation.

**Definition A.1.** *A finite hypergroup $G$ is a finite set with an involution $G \ni x \mapsto \overline{x} \in G$ such that $\mathbb{R}G$ is an associative algebra with the product*

$$xy = \sum_{z \in G} N_{xy}^z z, \qquad N_{xy}^z \in \mathbb{R}_{\geq 0}, \tag{A.1}$$

*where i) the unit is given by an element $e \in G$, ii) $\overline{xy} = \bar{y}\bar{x}$ where the involution is extended to $\mathbb{R}G$ by linearity, and iii) $N_{xy}^e \neq 0$ if and only if $x = \overline{y}$.*

**Definition A.2.** *Given a hypergroup $G$, let $\tilde{G}$ be a copy of $G$ with an element $\tilde{x} \in \tilde{G}$ for each $x \in G$. We choose a set of non-negative numbers $c_x$, and formally define $\tilde{x} = x/c_x$. Then $\tilde{G}$ becomes a hypergroup with the structure constants*

$$\tilde{N}_{\tilde{x}\tilde{y}}^{\tilde{z}} := \frac{c_z}{c_x c_y} N_{xy}^z. \tag{A.2}$$

*The hypergroups $G$ and $\tilde{G}$ can be considered as essentially the same. We say that $\tilde{G}$ is obtained from $G$ by a rescaling.*

*Remark* A.3. In the hypergroup literature, the standard normalization is to take $\sum_{\tilde{z}} \tilde{N}_{\tilde{x}\tilde{y}}^{\tilde{z}} = 1$. Another convention common in the literature is to take $\tilde{N}_{\tilde{x}\bar{\tilde{x}}}^{\tilde{e}} = 1$. Such rescalings can always be performed, but the proof of this fact (Props. A.28, A.29) is not immediate.

**Definition A.4.** *We call a hypergroup $G$ "strict" if $N_{x\overline{x}}^e = 1$ for all $x \in G$.*

*Remark* A.5. This usage of the adjective *strict* is not common in the literature. It is introduced here simply for the sake of exposition.

**Definition A.6.** *A finite hypergroup $G$ for which $N_{xy}^z$ are all integers is called a fusion algebra.*

*Remark* A.7. In the literature, strict fusion algebras in the sense above are often simply called fusion algebras. They are also called based rings, e.g. in [22].

*Example* A.8. A finite group $G$ determines a strict fusion algebra.

*Example* A.9. Given a finite group $G$, the elements $c([g]) = \sum_{h \sim g} h \in \mathbb{R}[G]$ generate a commutative subalgebra of $\mathbb{R}[G]$. Since the structure constants are integers, we see that the set $\mathrm{Conj}(G)$ of conjugacy classes forms a commutative fusion algebra. The algebra is not always strict in this normalization though, since $N_{[g][g^{-1}]}^{[e]} = \#[g]$ which can be larger than 1 when $G$ is non-Abelian. We can rescale the hypergroup by $c_g = (\#[g])^{1/2}$ to make it into a strict hypergroup, but then it is not necessarily a fusion algebra, since the structure constants are not necessarily integers.

*Example* A.10. Given a finite group $G$, the set $\mathrm{Rep}(G)$ of irreducible representations is also a fusion algebra, where the product is given by the tensor product. It is automatically strict, since $R \otimes \bar{R}$ contains the identity representation exactly once.

*Example* A.11. More generally, isomorphism classes of simple objects of a fusion category also form a finite fusion algebra. It is automatically strict.

*Example* A.12. A finite strict hypergroup structure on a two-element set $\{e, x\}$ is specified by $x^2 = e + nx$, where $n$ is a non-negative number. It is a fusion algebra when $n$ is a non-negative integer. When $n = 0$ or $n = 1$, it can be realized by a fusion category. In fact there are two distinct fusion categories for each $n = 0$ and $n = 1$ [35]. It is also known that there are no fusion categories realizing fusion algebras given by $x^2 = e + nx$ with $n \geq 2$ [36]. Therefore one needs to be careful about the distinction between hypergroups, fusion algebras, and fusion categories.

**Notation A.13.** *We define $w_x := N_{x\bar{x}}^e$. For $a = \sum_{x \in G} a_x x \in \mathbb{R}G$ with $a_x \in \mathbb{R}$, we write $u(a) = a_e$. We also define $x \prec a$ to mean $a_x \neq 0$.*

**Proposition A.14.** *We have $N_{xy}^z = u(xy\bar{z})/w_z$ and $N_{y\bar{z}}^{\bar{x}} = u(xy\bar{z})/w_x$. In particular, $z \prec xy$ is equivalent to $\bar{x} \prec y\bar{z}$.*

*Proof.* Immediate. □

*Remark* A.15. There are many other similar formulas, which will not be listed in full here but will be used implicitly below.

**Proposition A.16.** *For any $x, y \in G$, we have $xy \neq 0$.*

*Proof.* From Definition A.1, we have $e \prec \bar{x}x$ and $e \prec y\bar{y}$. Therefore $e \prec \bar{x}xy\bar{y}$. Therefore $xy \neq 0$. □

**Proposition A.17.** *For any $x, y \in G$, there is a $z \in G$ and $z' \in G$ such that $y \prec zx$ and $y \prec xz'$.*

*Proof.* We know that $y \prec zx$ is equivalent to $\bar{z} \prec x\bar{y}$. Since $x\bar{y}$ is nonzero via Prop. A.16, there is at least one such $z$. The other statement can be proved similarly. □

To define the quantum dimension and derive its properties, we use the following:

**Theorem A.18** (The Perron-Frobenius theorem). *Consider an $N \times N$ matrix $M_{ij}$ with non-negative entries. Then the following statements hold:*

- *Its maximal eigenvalue $\lambda(M)$ is non-negative.*

- *For this eigenvalue $\lambda(M)$, there is an eigenvector whose entries are all non-negative.*

- *If an eigenvector has all entries positive, its eigenvalue is $\lambda(M)$.*

- *If the entries $M_{ij}$ are all strictly positive, $\lambda(M)$ is non-degenerate and is positive.*

*We call such an eigenvalue/eigenvector the Perron-Frobenius eigenvalue/eigenvector of $M$.*

*Proof.* This is well-known; see e.g. [22, Sec. 3.2]. □

**Definition A.19.** *We denote the left and right multiplications by $x \in G$ by*

$$\ell_x : a \mapsto xa, \qquad r_x : a \mapsto ax. \tag{A.3}$$

*We denote the left and right multiplication by $\sum_{x \in G} x$ by*

$$\ell_G : a \mapsto (\sum_{x \in G} x)a, \qquad r_G : a \mapsto a(\sum_{x \in G} x). \tag{A.4}$$

*We regard them as linear operators on $\mathbb{R}G$.*

**Proposition A.20.** *For $R \in \mathbb{R}G$, the following four conditions are equivalent, and define $R$ as a vector with all positive entries, uniquely up to a positive scalar multiple:*

1. *$R$ is a common Perron-Frobenius eigenvector for all the right multiplications $r_x$.*

2. *$R$ is a Perron-Frobenius eigenvector for $r_G$.*

3. *$R$ is a common Perron-Frobenius eigenvector for all the left multiplications $\ell_x$.*

4. *$R$ is a Perron-Frobenius eigenvector for $\ell_G$.*

*Proof.* We prove this statement by showing that 1) implies 2), 2) implies 3), 3) implies 4), and 4) implies 1). That $R$ is unique up to a positive scalar multiple and that $R$ has all positive entries will be proved along the way.

1) to 2): Immediate.

2) to 3): The linear map $r_G$ as a matrix has strictly positive entries, thanks to Prop. A.17. Therefore, $R$ is unique up to scalar multiplication and its entries are all positive. Now we note that $xR$ is another eigenvector of $r_G$, whose entries are also all positive. From the uniqueness of the Perron-Frobenius eigenvector of $r_G$ up to scalar multiplication, we see that $xR \propto R$, meaning that $R$ is a Perron-Frobenius eigenvector of $\ell_x$.

3) to 4): Immediate.

4) to 1): Replace the role of the left and the right in the proof of 2) to 3). □

**Definition A.21.** *For $x \in G$, we define $d_x$ to be the Perron-Frobenius eigenvalue of the left multiplication $\ell_x$. We extend $d$ from $G$ to $\mathbb{R}G$ by linearity. We call $d_a$ the quantum dimension of $a$.*

**Proposition A.22.** *$d$ is a ring homomorphism $\mathbb{R}G \to \mathbb{R}$, i.e. $d_{xy} = d_x d_y$. Equivalently, $d_x d_y = \sum_{z \in G} N_{xy}^z d_z$.*

*Proof.* By definition $xR = d_x R$. From this it follows that $d_{xy}R = xyR = d_x d_y R$. □

**Proposition A.23.** *$d_x$ is also the Perron-Frobenius eigenvalue of the right multiplication $a \mapsto ax$ on $\mathbb{R}G$.*

*Proof.* We have already shown that $Rx = e_x R$ for some $e_x \in \mathbb{R}$. Taking the quantum dimension of both sides, we see that $d_R d_x = e_x d_R$. Therefore $e_x = d_x$. □

**Proposition A.24.** $d_x = d_{\bar{x}}$ .

*Proof.* This follows from the fact that the matrix of left multiplication by $x$ is equal to the matrix of right multiplication by $\bar{x}$. □

We can actually write the explicit form of the important element $R$ in Prop. A.20:

**Definition A.25.** *For a hypergroup G, we define its Haar element by*

$$R_G := \frac{\sum_{x \in G} d_x x / w_x}{\sum_{x \in G} d_x^2 / w_x} . \tag{A.5}$$

*Here the denominator is included to normalize $d_{R_G} = 1$.*

**Theorem A.26.** *We have $R_G y = y R_G = d_y R_G$. In particular, $R_G^2 = R_G$.*

*Proof.* We have

$$\left(\sum_{x \in G} d_x x / w_x\right) y = \sum_{x,z \in G} d_x N_{xy}^z / w_x z = \sum_{x,z \in G} d_x N_{y\bar{z}}^{\bar{x}} / w_z z = d_y \left(\sum_{z \in G} d_z z / w_z\right), \tag{A.6}$$

and therefore $R_G y = d_y R_G$. This means that $R_G$ satisfies the condition i) of Prop. A.20, from which $y R_G = d_y R_G$ also follows. □

**Proposition A.27.** $w_x = w_{\bar{x}}$ .

*Proof.* We consider the element

$$R_G' := \frac{\sum_{x \in G} d_x x / w_{\bar{x}}}{\sum_{x \in G} d_x^2 / w_{\bar{x}}} . \tag{A.7}$$

We can show that $y R_G' = d_y R_G'$, since

$$y\left(\sum_{x \in G} d_x x / w_{\bar{x}}\right) = \sum_{x,z \in G} d_x N_{yx}^z / w_{\bar{x}} z = \sum_{x,z \in G} d_x N_{\bar{z}y}^{\bar{x}} / w_{\bar{z}} z = d_y \left(\sum_{z \in G} d_z z / w_{\bar{z}}\right). \tag{A.8}$$

From Prop. A.20, we see $R_G = R_G'$, from which we see that $w_x = w_{\bar{x}}$. □

**Proposition A.28.** *We can rescale a given hypergroup G so that $\sum_z N_{xy}^z = 1$ for all $x$ and $y$. This is the standard convention in the hypergroup literature.*

*Proof.* We use $c_x = d_x$ in Definition A.2 and set $\tilde{x} = x / c_x$. Then $d_{\tilde{x}} = 1$. Evaluating the quantum dimension of both sides of $\tilde{x}\tilde{y} = \sum_{\tilde{z}} \tilde{N}_{\tilde{x}\tilde{y}}^{\tilde{z}} \tilde{z}$, one obtains $\sum_{\tilde{z}} \tilde{N}_{\tilde{x}\tilde{y}}^{\tilde{z}} = 1$. □

**Proposition A.29.** *We can rescale a given hypergroup G so that it is strict, i.e. $N_{x\bar{x}}^e = 1$. This is the standard convention in the fusion category literature.*

*Proof.* Use $c_x = \sqrt{w_x}$ in Definition A.2 and set $\tilde{x} = x / c_x$. Then

$$\tilde{N}_{\tilde{x}\tilde{\bar{x}}}^{\tilde{e}} = \frac{1}{\sqrt{w_x}\sqrt{w_{\bar{x}}}} N_{x\bar{x}}^e = 1 , \tag{A.9}$$

where we have used Prop. A.27. □

We now move on to a discussion of subhypergroups and quotients. For simplicity of presentation, we assume that all hypergroups are rescaled to be strict.

**Definition A.30.** *A subset $H$ of a hypergroup $G$ is a subhypergroup if $H$ is closed under the involution and $\mathbb{R}H$ is closed under the product.*

**Definition A.31.** *For a subhypergroup $H$ of $G$, we define 'the double coset $\langle x \rangle$ containing $x$' via*

$$\langle x \rangle = \{y \in G \mid y \prec h x h' \text{ for some } h, h' \in H\}. \tag{A.10}$$

**Proposition A.32.** *The relation $y \sim_H x$ defined by $y \in \langle x \rangle$ is an equivalence relation.*

*Proof.* Reflexivity is obvious. For transitivity, say $y \prec h x h'$ and $z \prec \tilde{h} y \tilde{h}'$ for some $h, h', \tilde{h}, \tilde{h}' \in H$. We then have $z \prec (\tilde{h}h)x(h'\tilde{h}')$. For symmetry, say $y \prec h x h'$. This means that $u(\bar{y}hxh') \neq 0$. This implies $u(\bar{h}'\bar{y}hx) \neq 0$, which then implies $u(\bar{x}\bar{h}yh') \neq 0$, meaning that $x \prec \bar{h}yh'$. $\qquad \square$

**Definition A.33.** *We define $G//H := \{\langle x \rangle\}$, the partition of $G$ given by the equivalence relation $\sim_H$.*

We now introduce the structure of a hypergroup on $G//H$. Consider the linear map

$$e_H : a \mapsto R_H a R_H, \tag{A.11}$$

defined on $\mathbb{R}G$. As $R_H^2 = R_H$, $e_H(\mathbb{R}G)$ clearly forms an $\mathbb{R}$-algebra. Let us find an explicit basis. For this, note

$$\mathbb{R}G = \bigoplus_{\langle x \rangle \in G//H} V_{\langle x \rangle}, \qquad V_{\langle x \rangle} = \bigoplus_{y \in \langle x \rangle} \mathbb{R}y. \tag{A.12}$$

Now consider the linear map

$$e_H|_{V_{\langle x \rangle}} : a \mapsto R_H a R_H. \tag{A.13}$$

By the definition of the double coset $\langle x \rangle$, $e_H|_{V_{\langle x \rangle}}$ as a matrix has strictly positive entries. Therefore its Perron-Frobenius eigenvector is unique. As $(e_H)^2 = e_H$, the Perron-Frobenius eigenvalue is 1. Now, $R_H R_G R_H = R_G$. Therefore the Perron-Frobenius eigenvector of $e_H|_{V_{\langle x \rangle}}$ is $\propto \sum_{y \in \langle x \rangle} d_y y$.

**Definition A.34.** *For $\langle x \rangle \in G//H$, we define*

$$R_{\langle x \rangle} := \frac{\sum_{y \in \langle x \rangle} d_y y}{\left(\sum_{z \in H} d_z^2\right)^{1/2} \left(\sum_{y \in \langle x \rangle} d_y^2\right)^{1/2}}. \tag{A.14}$$

*This reduces to $R_H$ when $x = e$.*

**Proposition A.35.** *The set $\{R_{\langle x \rangle}\}$ for $\langle x \rangle \in G//H$ spans a subalgebra of $\mathbb{R}G$.*

*Proof.* This is immediate since this set spans $R_H(\mathbb{R}G)R_H$ by the discussions above. $\qquad \square$

**Theorem A.36.** *Given a strict hypergroup $G$ and a subhypergroup $H$, we define $N_{\langle x \rangle \langle y \rangle}^{\langle z \rangle}$ via*

$$R_{\langle x \rangle} R_{\langle y \rangle} = \sum_{\langle z \rangle \in G//H} N_{\langle x \rangle \langle y \rangle}^{\langle z \rangle} R_{\langle z \rangle}. \tag{A.15}$$

*This makes $G//H$ into a strict hypergroup, with the multiplication given by*

$$\langle x \rangle \langle y \rangle = \sum_{\langle z \rangle \in G//H} N_{\langle x \rangle \langle y \rangle}^{\langle z \rangle} \langle z \rangle. \tag{A.16}$$

*Proof.* The only remaining step is to show that $N^{\langle e \rangle}_{\langle x \rangle \langle \bar{x} \rangle} = 1$ and that $N^{\langle e \rangle}_{\langle x \rangle \langle y \rangle} \neq 0$ if and only if $y = \bar{x}$. The former is a simple computation done by computing $u(R_{\langle x \rangle} R_{\langle \bar{x} \rangle})$. To show the latter, assume that $N^{\langle e \rangle}_{\langle x \rangle \langle y \rangle} \neq 0$. Then $h \prec xh'y$ for some $h, h' \in H$, which means $u(\bar{h}xh'y) \neq 0$. This is equivalent to $y \prec \bar{h}'\bar{x}h$ and therefore $y \in \langle \bar{x} \rangle$. $\qquad\square$

*Remark* A.37. $G//H$ is not necessarily a strict fusion algebra even when $G$ and $H$ are. This is due to the fact that $N^{\langle z \rangle}_{\langle x \rangle \langle y \rangle}$ as determined above are not necessarily integers, even when $N^z_{xy}$ are.

Note that the construction of the quotient hypergroup $G//H$ was done without any assumption of normality for $H$. We will now study the effect of two types of normalities on $H$.

**Definition A.38.** *A subhypergroup $H$ of $G$ is called normal when $Hx = xH$ for all $x \in G$, where*

$$Hx := \{y \mid y \prec hx \text{ for some } h \in H\}, \qquad xH := \{y \mid y \prec xh \text{ for some } h \in H\}. \tag{A.17}$$

*A subhypergroup $H$ of $G$ is called supernormal when $xH\bar{x} \subset H$ for all $x \in G$, where*

$$xH\bar{x} := \{y \mid y \prec xh\bar{x} \text{ for some } h \in H\}. \tag{A.18}$$

*Remark* A.39. For a subgroup $H$ of a group $G$, being supernormal is equivalent to being normal. Also, $G/H$ when $H$ is normal agrees with $G//H$ as hypergroups.

**Proposition A.40.** *A supernormal subhypergroup $H$ of a hypergroup $G$ is normal.*

*Proof.* Suppose $y \prec hx$ for some $h \in H$. It suffices to show that $y \prec xh'$ for some $h' \in H$. To do so, note that supernormality implies $x\bar{x} \in \mathbb{R}H$, and therefore $y \prec (x\bar{x})hx = x(\bar{x}hx)$. From $\bar{x}Hx \subset H$, we see that we can pick an $h' \in H$ such that $y \prec xh'$. $\qquad\square$

**Proposition A.41.** *For a normal subhypergroup $H$, we have $Hx = xH = HxH$.*

*Proof.* Immediate. $\qquad\square$

*Remark* A.42. This means that for a normal subhypergroup, one can introduce a hypergroup structure on the left coset as in the group case, since it is just a special case of the quotient hypergroup structure on the double coset.

The condition of normality appears naturally in relation to morphisms between hypergroups.

**Definition A.43.** *Given two hypergroups $L$ and $K$, a map $\phi : L \to K$ is a morphism if it preserves the involution and if it extends to an algebra homomorphism $\phi : \mathbb{R}L \to \mathbb{R}K$. The image of a morphism $\phi : L \to K$ is simply the image as the map between two sets. The kernel of a morphism $\phi : L \to K$ is the inverse image of $e \in K$.*

**Proposition A.44.** *The image of a morphism is a hypergroup.*

*Proof.* Immediate. $\qquad\square$

**Proposition A.45.** *The kernel $H$ of a morphism $\phi : L \to K$ is a normal subhypergroup of $L$.*

*Proof.* Let us show $xH = \phi^{-1}(\phi(x))$. To show this, say $y \prec xh$ for some $h \in H$. Applying $\phi$, we have $\phi(y) \prec \phi(x)\phi(h) = \phi(x)$. Conversely, say $y \prec xh$ for no $h \in H$. This means $h \prec \bar{y}x$ for no $h \in H$. Therefore $\phi(\bar{y})\phi(x)$ does not contain $e$, and $\phi(x) \neq \phi(y)$. We conclude that $xH = \phi^{-1}(\phi(x))$. We can analogously show that $Hx = \phi^{-1}(\phi(x))$. Therefore $xH = Hx$. $\qquad\square$

*Remark* A.46. Summarizing, associated to a normal subhypergroup $H \subset G$, there is a short exact sequence

$$\{e\} \to H \to G \to G//H \to \{e\}, \tag{A.19}$$

of hypergroup morphisms.

**Proposition A.47.** *The quotient hypergroup $G//H$ is a group if and only if $H$ is supernormal.*

*Proof.* If $G//H$ is a group, then $R_{\langle x \rangle} R_{\langle \bar{x} \rangle} = R_H$. This requires that $x R_H \bar{x} \in \mathbb{R}H$, meaning that $x H \bar{x} \subset H$. So $H$ is supernormal.

Suppose conversely that $H$ is supernormal. Let us first show that $R_{\langle x \rangle} R_{\langle y \rangle} \propto R_{\langle z \rangle}$ for some $z$. For this, note that $R_{\langle x \rangle} \propto R_H x R_H$ and $R_{\langle y \rangle} \propto R_H y R_H$. It then suffices to show that if $z \prec xhy$ and $w \prec xh'y$ for $h, h' \in H$, we have $z \sim_H w$. To see this, note that $z \prec xhy$ implies $x \prec z\bar{y}\bar{h}$, meaning that $w \prec (z\bar{y}\bar{h})h'y = z(\bar{y}(\bar{h}h')y)$. Therefore $w \prec zh''$, where $h'' \in \bar{y}Hy \subset H$.

Next, to determine the proportionality constant, note that $R_{\langle x \rangle} R_{\langle \bar{x} \rangle} = R_H$ thanks to our choice of normalizations. This means that $d_{\langle x \rangle} = 1$ for all $x$, which in turn implies that $R_{\langle x \rangle} R_{\langle y \rangle} = R_{\langle z \rangle}$ for some $z$. Therefore we have found that $G//H$ is indeed a group. $\square$

*Remark* A.48. Note that in this case we have a hypergroup morphism $f : G \to X := G//H$ where $G$ is a hypergroup and $X$ is a group. If we expand the definition, this simply means that $N_{xy}^z \neq 0$ only if $f(x)f(y) = f(z)$. Therefore $X$ gives a grading of the hypergroup multiplication law and/or the fusion rule coefficients $N_{xy}^z$. It is natural to ask what the largest possible such grading $X$ is, or equivalently, to ask what the smallest possible supernormal subgroup $H$ is.

**Definition A.49.** *Let* $\mathrm{Com}(G) = \{h \mid h \prec x\bar{x} \text{ for some } x \in G\}$. *We then define* $\mathrm{Com}(G)^L$ *to be*

$$\mathrm{Com}(G)^L = \{y \mid y \prec x_1 x_2 \cdots x_L \text{ for some } x_1, x_2, \ldots, x_L \in \mathrm{Com}(G)\}. \tag{A.20}$$

**Proposition A.50.** *We have* $\mathrm{Com}(G)^L \subset \mathrm{Com}(G)^{L+1}$.

*Proof.* This follows from the fact that $e \in \mathrm{Com}(G)$. $\square$

**Definition A.51.** *The ascending chain*

$$\mathrm{Com}(G) \subset \mathrm{Com}(G)^2 \subset \cdots \mathrm{Com}(G)^L \subset \mathrm{Com}(G)^{L+1} \subset \cdots, \tag{A.21}$$

*eventually stabilizes, as they are subsets of a finite set $G$. We denote the limit by* $\mathrm{Com}(G)^\infty$.

**Proposition A.52.** $\mathrm{Com}(G)^\infty$ *is a supernormal subgroup of $G$.*

*Proof.* For finite $L$, whenever $h \in \mathrm{Com}(G)^L$, we have $xh\bar{x} = x\bar{x}h \in \mathrm{Com}(G)^{L+1}$, so $xH\bar{x} \subset \mathrm{Com}(G)^{L+1}$. Taking $L \to \infty$, we have $x \mathrm{Com}(G)^\infty \bar{x} \subset \mathrm{Com}(G)^\infty$. $\square$

**Proposition A.53.** *Any supernormal subgroup $H$ of $G$ contains* $\mathrm{Com}(G)^\infty$.

*Proof.* As $e \in H$ and $xH\bar{x} \subset H$, we have $\mathrm{Com}(G) \subset H$. As $H$ is a subhypergroup, it follows that $\mathrm{Com}(G)^\infty \subset H$. $\square$

**Definition A.54.** *We define the "groupification"* $\mathrm{Gr}[G]$ *of $G$ by* $\mathrm{Gr}[G] := G//\mathrm{Com}(G)^\infty$.

*Remark* A.55. From the discussions above, the groupification $\mathrm{Gr}[G]$ gives the finest grading $G \to \mathrm{Gr}[G]$ of a given hypergroup $G$. Possible gradings $G \to X$ of a strict fusion algebra $G$ by a group $X$ were discussed without introducing general hypergroups and their quotients in [22, Sec. 3.5]. Although the conclusion there is important and useful, the discussion seems rather *ad hoc*. The authors recalled the basic theory of hypergroups here since they thought that hypergroups might provide slightly more context for the problem at hand.

# B  Tree- and loop-level selection rules of non-invertible symmetries of WZW models

In this appendix we study the tree- and loop-level selection rules coming from non-invertible symmetries of WZW models based on $\mathfrak{g}_k$ affine Lie algebras. Before proceeding, we note that the WZW models themselves have the full $\mathfrak{g}_k$ affine symmetry, which includes the ordinary Lie algebra symmetry for $\mathfrak{g}$. The selection rules arising from these Lie algebras would be stronger than the selection rules coming solely from the fusion algebras associated to $\mathfrak{g}_k$. The aim of this appendix is simply to use these fusion algebras as concrete examples for which the techniques introduced in the main text can be applied.

Non-invertible symmetries of the diagonal $\mathfrak{g}_k$ WZW model are given by a modular tensor category, whose simple objects are irreducible integrable representations of $\mathfrak{g}_k$. As such, its closed string Hilbert space is organized into sectors labeled by $A(\mathfrak{g}_k) = P_+^k$, the positive weights of $\mathfrak{g}$ whose heights are less than $k$. $A(\mathfrak{g}_k)$ forms a commutative fusion algebra, to which we can apply our methods. In particular, $A(\mathfrak{g}_k)$ itself gives the tree-level selection rules, which reduces to a symmetry labeled by a finite Abelian group $\mathrm{Gr}[A(\mathfrak{g}_k)]$ at infinite loop level.

Our main interest will be in studying the "conjugate pair length" $\mathrm{cl}(A(\mathfrak{g}_k)) \in \mathbb{Z}_+$, which specifies the loop level at which the selection rules reduce to those of $\mathrm{Gr}[A(\mathfrak{g}_k)]$. As was explained in Sec. 2.3, conjugate pair length generalizes the notion of commutator length for a finite group. Concretely, we will find that $A(\mathfrak{su}(2)_k)$, the fusion algebra for the diagonal $\mathfrak{su}(2)_k$ WZW model, has conjugate pair length equal to 1. A similar statement holds for the fusion algebras of non-diagonal $\mathfrak{su}(2)_k$ WZW models, when the chiral algebra extends. On the other hand, we will see that the even part $A'$ of the fusion algebra $A(\mathfrak{su}(2)_{2k})$ has $\mathrm{cl}(A') = 2$. This even part is not a modular tensor category, and therefore cannot be directly used to label the closed-string sectors of a perturbative string theory, but nevertheless may be interesting in other contexts. Finally, using computational means we show that $\mathrm{cl}(A(\mathfrak{su}(N)_k)) = 1$ for $N, k \leq 7$.

## B.1  Affine characters and fusion rules

We begin by reviewing the basic WZW fusion rules; for a comprehensive review, we refer the reader to [37]. The primary fields in a $\mathfrak{g}_k$ WZW model are labeled by integral representations $\lambda$ of the $\mathfrak{g}_k$ affine Lie algebra at level $k$, which descends to a simple Lie algebra $\mathfrak{g}$. If we consider a process in which incoming integrable representations $\lambda$ and $\mu$ turn into an outgoing integrable representation $\nu$, then the fusion coefficients $\mathcal{N}_{\lambda\mu}^{(k)\nu}$ are defined by

$$\lambda \hat{\otimes} \mu = \bigoplus_{\sigma \in P_+^k} \mathcal{N}_{\lambda\mu}^{(k)\nu} \nu, \tag{B.1}$$

where $P_+^{(k)}$ is the affine Weyl chamber of $\mathfrak{g}$ at level k. For $\mathfrak{su}(N)_k$, an element $x \in P_+^{(k)}$ is simply a sequence of $N$ non-negative integers $[x_1, \ldots, x_N]$ with $x_1 + \cdots + x_N = k$. In particular, for $N = 2$ we have

$$P_+^{(k)} = \{[k, 0], [k-1, 1], [k-2, 2], \ldots, [0, k]\}. \tag{B.2}$$

The fusion coefficients $\mathcal{N}_{\lambda\mu}^{(k)\nu}$ can be computed from the modular $\mathcal{S}$ matrix using the Verlinde formula [38],

$$\mathcal{N}_{\lambda\mu}^{(k)\nu} = \sum_{\sigma \in P_+^k} \frac{\mathcal{S}_{\lambda\sigma} \mathcal{S}_{\mu\sigma} \overline{\mathcal{S}}_{\nu\sigma}}{\mathcal{S}_{0\sigma}}. \tag{B.3}$$

For $\mathfrak{su}(N)_k$, they can alternatively be computed via a combinatorial algorithm based on Young tableaux. Roughly speaking, this proceeds in two steps,

Table 1: Modular invariants of $\mathfrak{su}(2)_k$. For more details, see [37].

$$
\begin{array}{lll}
A_{k+1}: & \sum_{n=0}^{k} |\chi_n|^2 & (\forall\, k) \\
D_{2\ell+2}: & \sum_{n=0}^{\ell-1} |\chi_{2n} + \chi_{4\ell-2n}|^2 + 2|\chi_{2\ell}|^2 & (k = 4\ell) \\
D_{2\ell+1}: & \sum_{n=0}^{2\ell-1} |\chi_{2n}|^2 + |\chi_{2\ell-1}|^2 + \sum_{n=0}^{\ell-2}(\chi_{2n+1}\bar{\chi}_{4\ell-2n-3} + \text{c.c.}) & (k = 4\ell-2) \\
E_6: & |\chi_0 + \chi_6|^2 + |\chi_3 + \chi_7|^2 + |\chi_4 + \chi_{10}|^2 & (k = 10) \\
E_7: & |\chi_0 + \chi_{16}|^2 + |\chi_4 + \chi_{12}|^2 + |\chi_6 + \chi_{10}|^0 + |\chi_8|^2 + (\chi_8(\bar{\chi}_2 + \bar{\chi}_{14}) + \text{c.c.}) & (k = 16) \\
E_8: & |\chi_0 + \chi_{10} + \chi_{18} + \chi_{28}|^2 + |\chi_6 + \chi_{12} + \chi_{16} + \chi_{22}|^2 & (k = 28)
\end{array}
$$

- First, we determine the fusion coefficients $\mathcal{N}_{\lambda\mu}^{\nu}$ of the non-affine Lie algebra $\mathfrak{g}$. These can be obtained by e.g. the Littlewood-Richardson algorithm [39].

- Next we perform the affinization. Concretely, given a specific level $k$, we put the non-affine coefficients into the Kac-Walton formula [40, 41],

$$
\mathcal{N}_{\lambda\mu}^{(k)\nu} = \sum_{w \in \hat{W},\ w\cdot\nu \in P_+} \mathcal{N}_{\lambda\mu}^{w\cdot\nu} \epsilon(w), \tag{B.4}
$$

where $\hat{W}$ is the affine Weyl group of $\mathfrak{su}(N)_k$. Later we will see that this step amounts to doing a truncation on the tensor product coefficients of $\mathfrak{su}(N)$ representations, and that such a truncation always becomes trivial under the large $k$ limit $\mathcal{N}_{\lambda\mu}^{(\infty)\nu} = \mathcal{N}_{\lambda\mu}^{\nu}$.

In some cases it is possible to write a closed form expression for $\mathcal{N}_{\lambda\mu}^{(k)\nu}$, as is the case for example for $\mathfrak{su}(2)_k$ [42, 43],

$$
\mathcal{N}_{\lambda\mu}^{(k)\nu} = \begin{cases} 1, & |\lambda - \mu| \leq \nu \leq \min\{\lambda + \mu, 2k - \lambda - \mu\} \text{ and } \lambda + \mu + \nu = 0 \mod 2, \\ 0, & \text{otherwise.} \end{cases} \tag{B.5}
$$

For a general $\mathfrak{g}_k$ WZW model, the modular-invariant partition function can be written in terms of holomorphic and anti-holomorphic characters $\chi_\lambda$, $\bar{\chi}_{\lambda'}$, together with a pairing matrix $\mathcal{M}$,

$$
Z = \sum_{\lambda, \lambda'} \mathcal{M}_{\lambda\lambda'} \chi_\lambda \bar{\chi}_{\lambda'}. \tag{B.6}
$$

When $\mathcal{M}$ is the identity matrix, the modular invariant is referred to as diagonal. Whenever we discuss a $\mathfrak{g}_k$ WZW model without specifying the partition function, we are implicitly working with the diagonal invariant. In such cases, each characters appear exactly once, and all fields are spinless, meaning the holomorphic and anti-holomorphic conformal dimensions are the same, i.e. $h = \bar{h}$.

It is also possible to construct WZW theories with off-diagonal modular invariants. In the case of $\mathfrak{g} = \mathfrak{su}(2)$ there are two infinite families of non-diagonal D-type invariants, as well as a series of exceptional non-diagonal invariants, with corresponding partition functions given in Table 1. As indicated there, the existence of these invariants depends on the particular value of the level $k$. For example, for the $\mathfrak{su}(2)_4$ theory, there is a single off-diagonal modular invariant, given by

$$
Z = |\chi_0|^2 + |\chi_4|^2 + 2|\chi_2|^2 + \chi_0\bar{\chi}_4 + \chi_4\bar{\chi}_0. \tag{B.7}
$$

This corresponds to the invariant of type $D_4$. This off-diagonal modular invariant can be re-written as $|\chi_0 + \chi_4|^2 + 2|\chi_2|^2$, with the following interpretation: under the $\mathbb{Z}_2$ outer-automorphism folding, $\chi_0$ and $\chi_4$ combines into a new state, while the invariant $\chi_2$ splits into two separate states.

This behavior persists for all $D_{2\ell+2}$ type theories for $k = 4\ell$. The fusion rules in these cases are given as follows [44]. First, the irreducible modules are given by $\langle n \rangle := [2n, 4\ell - 2n] + [4\ell - 2n, 2n]$ for $0 \le n < \ell$, which are the invariant combinations of characters under automorphism, together with $\langle \ell \rangle_\pm$, which are the pair arising from the middle characters $[2\ell, 2\ell]$ which are invariant under the automorphism. The fusion rules are then,

$$
\begin{aligned}
\langle n_1 \rangle \langle n_2 \rangle &= \sum_{j=|n_1-n_2|}^{\max(|n_1+n_2|, 8\ell - |n_1+n_2|)} \langle j \rangle, \\
\langle n \rangle \langle l \rangle &= \sum_{j=l-n,\ j \ne l}^{l+n} \langle j \rangle + x_n \langle l \rangle_+ + (1 - x_n) \langle l \rangle_-, \\
\langle l \rangle_\pm \langle l \rangle_\pm &= \sum_{n=0}^{l-1} x_{n+l} \langle n \rangle + \langle x_\pm \rangle, \\
\langle l \rangle_\pm \langle l \rangle_\mp &= \sum_{n=0}^{l-1} (1 - x_{n+l}) \langle n \rangle,
\end{aligned}
\tag{B.8}
$$

where $x_n = 1$ for $n$ odd and $0$ for $n$ even, with $x_{n+l}$ being understood similarly. We have also introduced $\langle 2l - n \rangle := \langle n \rangle$ and $\langle l \rangle := \langle l \rangle_+ + \langle l \rangle_-$ for notational uniformity.

One way to generate diagonal invariants is to consider conformal embeddings $\mathfrak{su}(2)_k \subset \mathfrak{g}_{k'}$, for which the diagonal invariants of $\mathfrak{g}_{k'}$ induce non-diagonal invariants of $\mathfrak{su}(2)_k$. For example, the case of $D_4$ mentioned above follows from the conformal embedding $\mathfrak{su}(2)_4 \subset \mathfrak{su}(3)_1$. A similar thing can be done using the conformal embeddings $\mathfrak{su}(2)_{10} \subset \mathfrak{sp}(2)_1$ and $\mathfrak{su}(2)_{28} \subset (G_2)_1$, which give rise to the $E_6$ and $E_8$ invariants, respectively. The fusion rules of the $E_6$ and $E_8$ modular invariants can then be obtained by recalling that the fusion algebras of $(G_2)_1$ and $\mathfrak{sp}(2)_1$ are Fibonacci and Ising, respectively.

## B.2   Symmetries at infinite loop order

We next analyze the symmetries of the diagonal $\mathfrak{su}(N)_k$ WZW model at arbitrary loop order. As discussed in the main text, this is given by the finite group $\mathrm{Gr}[A] = A // \mathrm{Com}(A)^\infty$. Let us first show that for $A = A(\mathfrak{g}_k)$, the group $\mathrm{Gr}[A]$ contains the center $Z(G)$ of $G$, where $G$ is the simply-connected group with Lie algebra $\mathfrak{g}$. To see this, note that the center $Z(G)$ acts on all possible representations of the affine Lie algebra, so a conjugate pair of representations has opposite center charges,

$$
c(\lambda) = -c(\bar\lambda) \in Z(G),
\tag{B.9}
$$

where $c(\lambda)$ is the center charge of the irreducible representation $\lambda \in P_+^k$. Therefore, the fusion product of $\lambda$ and $\bar\lambda$ has trivial center charge[17]

$$
c(\lambda \cdot \bar\lambda) = 0 \in Z(G),
\tag{B.11}
$$

and therefore

$$
x \prec \lambda \cdot \bar\lambda \quad \Rightarrow \quad c(x) = 0 \in Z(G).
\tag{B.12}
$$

From this it follows that

$$
\mathrm{Com}(A)^\infty \subset \{x \mid c(x) = 0 \in Z(G)\},
\tag{B.13}
$$

---

[17]More generally, the triple fusion coefficients are zero for those triplets whose center charge do not sum properly:

$$
c(\lambda) + c(\mu) \ne c(\nu) \in Z(G) \quad \Rightarrow \quad N_{\lambda\mu}^{(k)\nu} = 0.
\tag{B.10}
$$

which then means that

$$\text{Gr}[A] = A//\text{Com}(A)^\infty \supset A//\{x \mid c(x) = 0 \in Z(G)\} = Z(G). \tag{B.14}$$

In particular, this means that for $\mathfrak{su}(N)_k$ WZW models, the all-loop symmetry will at least contain $\mathbb{Z}_N$. Conversely, if we are able to show that every irreducible representation with charge $0 \in Z(G)$ appears in $\text{Com}(A)^\infty$, then we can conclude that $\text{Gr}[A] = Z(G)$. We will do this for $\mathfrak{su}(2)_k$ below.

Incidentally, note that the conclusion (B.13) can also be understood from the statement (53) quoted in the main text. There, we noted that a mathematical theorem guarantees that $\text{Gr}[A]$ equals the group $\text{Inv}(A)$ of invertible objects of $A$. For $\mathfrak{su}(N)_k$, it is easy to find a $\mathbb{Z}_N$ subgroup of $\text{Inv}(A)$. Indeed, the irreducible representations $[k,0,\dots,0]$, $[0,k,\dots,0]$, $\dots$, $[0,0,\dots,k]$ of $\mathfrak{su}(N)_k$ can be checked to form a $\mathbb{Z}_N$ subgroup, and it turns out that these elements give *all* the invertible elements in the $\mathfrak{su}(N)_k$ fusion category, as shown in [45].

### B.3 Conjugate pair lengths

We next study the conjugate pair length $\text{cl}(A)$ of the fusion algebras associated to WZW models, starting with the following result.

**Proposition B.1.** *For any $k$, the fusion algebra $A = A(\mathfrak{su}(2)_k)$ for the diagonal $\mathfrak{su}(2)_k$ WZW model satisfies*

$$\text{Gr}[A] = \mathbb{Z}_2, \qquad \text{and} \qquad \text{cl}(A) = 1. \tag{B.15}$$

*Proof.* This is immediate upon noting that the full set of $\mathbb{Z}_2$-symmetric elements is

$$[k-j,j], \qquad 0 \le j \le \left\lfloor \frac{k}{2} \right\rfloor, \tag{B.16}$$

with $j$ taking integer values, together with the fact that the non-$\mathbb{Z}_2$-symmetric element with $j = \frac{1}{2}\lfloor \frac{k}{2} \rfloor$ satisfies

$$[k-j,j] \cdot [k-j,j] = \sum_{\ell=0}^{\lfloor \frac{k}{2} \rfloor} [k-\ell,\ell]. \tag{B.17}$$

Because all of the $\mathfrak{su}(2)_k$ affine weights with center charge $0 \in \mathbb{Z}_2$ appear in this fusion product, the results of the previous section imply that $\text{Gr}[A] = \mathbb{Z}_2$. Since this occurs already at one loop, we conclude that $\text{cl}(A) = 1$. $\qquad\square$

Similar statements hold for the diagonal invariants of $\mathfrak{su}(3)_k$ for arbitrary $k$. The proof proceeds by using the explicit form of the fusion rules in [46], though we do not describe it here. In the more general case of $\mathfrak{su}(N)_k$, we have checked via explicit computation in SAGEMATH that the following holds,

$$\text{Gr}[(A(\mathfrak{su}(N)_k)] = \mathbb{Z}_N, \qquad \text{and} \qquad \text{cl}(A(\mathfrak{su}(N)_k)) = 1, \quad \text{for } N, k \le 7. \tag{B.18}$$

It would be interesting to understand whether this pattern persists more generally.

Next, let us take $k = 4\ell$ consider the fusion algebra $D_{\mathfrak{su}_k}$ given in (B.8). In this case we have the following result,

**Proposition B.2.** *We have*

$$\text{Gr}[D_{\mathfrak{su}_k}] = \mathbb{Z}_1, \qquad \text{and} \qquad \text{cl}(D_{\mathfrak{su}(2)_k}) = 1. \tag{B.19}$$

*Proof.* This follows immediately by considering the fusion of the self-conjugate weight $[2\ell-2]$ with itself, which by (B.8) includes all basis elements. $\qquad\square$

The cases of the $E_6$ and $E_8$ modular invariants are simpler than the cases discussed above, since the fusion algebras in those cases are given simply by the Ising and Fibonacci fusion algebras, respectively.

Finally, assume that $k$ is even, and consider the $\mathbb{Z}_2$-even part $A'_{\mathfrak{su}_k} \subset A(\mathfrak{su}_k)$ of the $\mathfrak{su}(2)_k$ fusion algebra, given by

$$A'_{\mathfrak{su}_k} = \{[k,0],\ [k-2,2],\ [k-4,4],\ ,\ldots,[0,k]\}\,. \tag{B.20}$$

This is a subalgebra of the fusion algebra $A(\mathfrak{su}_k)$, and in fact gives a fusion subcategory of the modular tensor category $\mathcal{A}_{\mathfrak{su}_k}$. This subcategory is not itself modular, though.[18] We then have the following results,

**Proposition B.3.** *The $\mathbb{Z}_2$-even part $A'_{\mathfrak{su}_k} \subset A(\mathfrak{su}_k)$ of the $\mathfrak{su}(2)_k$ fusion algebra satisfies,*

$$\mathrm{Gr}[A'_{\mathfrak{su}_k}] = \mathbb{Z}_2\,, \qquad \text{and} \qquad \mathrm{cl}(A'_{\mathfrak{su}_k}) = \begin{cases} 1\,, & \frac{k}{2}\,\text{even}\,, \\ 2\,, & \frac{k}{2}\,\text{odd}\,. \end{cases} \tag{B.21}$$

*Proof.* For $k/2$ even, the affine weight $[k,k]$ is in $A'_{\mathfrak{su}_k}$. The proof then proceeds in the usual fashion: we note that $[k,k]$ is self-conjugate, and a short computation shows that the product $[k,k]\cdot[k,k]$ contains all generators.

On the other hand, for $k/2$ odd, we see that a single conjugate pair can produce all weights ranging from $[2k,0]$ to $[2,2k-2]$, but not $[0,2k]$. These cases must thus have conjugate pair length greater than one. Proceeding to two loops, we note that

$$[k+1,k-1],\ [k-1,k+1] \prec [k+1,k-1]\cdot[k+1,k-1], \tag{B.22}$$

while

$$[0,2k] \prec [k+1,k-1]\cdot[k-1,k+1]\,. \tag{B.23}$$

We have thus generated all basis elements at two loops, and hence $\mathrm{cl}(A'_{\mathfrak{su}_k}) = 2$. $\qquad\square$

# C  Low-rank fusion algebras with conjugate pair length greater than one

All of the explicit examples of hypergroups encountered in the main text have conjugate pair length $\mathrm{cl}(A)$ equal to one; in other words, $\mathrm{Com}(A) = \mathrm{Com}(A)^\infty$. As a result, the selection rules studied in the main text all reduce to their all-loop forms already at one-loop. It is interesting to ask for examples of hypergroups with $\mathrm{cl}(A) > 1$. In this appendix, we restrict to fusion algebras of multiplicity 1 (i.e. those whose fusion coefficients $N^z_{x,y}$ are 0 or 1), whose classification can be found for rank $r \leq 9$ in e.g. [48], and list all examples with $\mathrm{cl}(A) > 1$ for $r \leq 7$. In fact, we will see that all such examples have $\mathrm{cl}(A) = 2$. Achieving higher values of $\mathrm{cl}(A)$ seems to require proceeding to higher rank or multiplicity. Let us also mention that none of the examples below can be realized by modular tensor categories [48]. In fact, most of them cannot be realized by standard fusion categories either [49]. We will indicate when they can be realized as such. A brief summary of our results can be found in Table 2.

---

[18]It is, however, super-modular, in the sense that there is a single invertible order-2 element (a 'fermion') which has trivial braiding with everything else [47]. In this case, the 'fermion' is the irreducible representation $[0,2k]$.

Table 2: At each rank, we list the number of distinct multiplicity-1 fusion algebras, the number which are non-Abelian, the number which are categorifiable, the number which have $\mathrm{cl}(A) > 1$ (in all cases, $\mathrm{cl}(A) = 2$), and the number which both have $\mathrm{cl}(A) > 1$ and are categorifiable. There is only one case which is both non-Abelian and has $\mathrm{cl}(A) > 1$, namely the Haagerup-Izumi $\mathrm{HI}(\mathbb{Z}_3)$ fusion algebra at rank-6. This case is also categorifiable.

| Rank | Algebras | Non-Abelian | Categorifiable | $\mathrm{cl}(A) > 1$ | $\mathrm{cl}(A) > 1$ and categorifiable |
|---|---|---|---|---|---|
| 3 | 4 | 0 | 4 | 0 | 0 |
| 4 | 10 | 0 | 9 | 2 | 2 |
| 5 | 16 | 0 | 10 | 1 | 0 |
| 6 | 39 | 2 | 21 | 11 | 2 |
| 7 | 43 | 3 | 12 | 5 | 0 |

## C.1 Rank 4

We begin at rank 4, where there are two multiplicity-1 fusion algebras with conjugate pair length greater than 1 (in particular, 2). Their fusion rules are contained in the following tables, where the element 1 represents the identity and $2, 3, 4$ represent the non-trivial objects,

$$
\begin{array}{c|cccc}
 & 1 & 2 & 3 & 4 \\
\hline
1 & 1 & 2 & 3 & 4 \\
2 & 2 & 2+3+4 & 1+2+3 & 3 \\
3 & 3 & 1+2+3 & 2+3+4 & 2 \\
4 & 4 & 3 & 2 & 1
\end{array}
\qquad
\begin{array}{c|cccc}
 & 1 & 2 & 3 & 4 \\
\hline
1 & 1 & 2 & 3 & 4 \\
2 & 2 & 1 & 4 & 3 \\
3 & 3 & 4 & 1+3+4 & 2+3+4 \\
4 & 4 & 3 & 2+3+4 & 1+3+4
\end{array}
\tag{C.1}
$$

In both cases $\mathrm{Com}(A)^2 = \mathrm{Com}(A)^\infty = \{1, 2, 3, 4\}$. The second of these fusion rules is recognized as the fusion ring for $\mathrm{PSU}(2)_6$, discussed previously in App. B, or equivalently for the Haagerup-Izumi category $\mathrm{HI}(\mathbb{Z}_2)$. These fusion rules can be realized by a fusion category, and also admit a braiding, but they cannot be realized by a modular tensor category. The total quantum dimensions of both cases are $\mathcal{D}^2 \approx 13.6569$.

## C.2 Rank 5

At rank 5, there is only a single multiplicity-1 fusion algebra with conjugate pair length greater than 1 (in particular, 2). Its fusion rules are given in the following table,

$$
\begin{array}{c|ccccc}
 & 1 & 2 & 3 & 4 & 5 \\
\hline
1 & 1 & 2 & 3 & 4 & 5 \\
2 & 2 & 1 & 5 & 4 & 3 \\
3 & 3 & 5 & 1+4+5 & 3+5 & 2+3+4 \\
4 & 4 & 4 & 3+5 & 1+2+4 & 3+5 \\
5 & 5 & 3 & 2+3+4 & 3+5 & 1+4+5
\end{array}
\tag{C.2}
$$

with $\mathrm{Com}(A)^2 = \mathrm{Com}(A)^\infty = \{1, 2, 3, 4, 5\}$. These fusion rules cannot be realized by a fusion category. The total quantum dimension is $\mathcal{D}^2 \approx 16.6056$.

## C.3 Rank 6

At rank 6, there are eleven multiplicity-1 fusion algebras with conjugate pair length greater than 1 (in particular, 2). Of them, only two can be realized by fusion categories. These two sets of fusion rules are given by the following tables,

|   | 1 | 2 | 3 | 4 | 5 | 6 |
|---|---|---|---|---|---|---|
| 1 | 1 | 2 | 3 | 4 | 5 | 6 |
| 2 | 2 | 1 | 6 | 5 | 4 | 3 |
| 3 | 3 | 6 | $1+5+6$ | $4+5+6$ | $3+4+5$ | $2+3+4$ |
| 4 | 4 | 5 | $4+5+6$ | $1+3+4+5+6$ | $2+3+4+5+6$ | $3+4+5$ |
| 5 | 5 | 4 | $3+4+5$ | $2+3+4+5+6$ | $1+3+4+5+6$ | $4+5+6$ |
| 6 | 6 | 3 | $2+3+4$ | $3+4+5$ | $4+5+6$ | $1+5+6$ |

$$(C.3)$$

|   | 1 | 2 | 3 | 4 | 5 | 6 |
|---|---|---|---|---|---|---|
| 1 | 1 | 2 | 3 | 4 | 5 | 6 |
| 2 | 2 | 3 | 1 | 6 | 4 | 5 |
| 3 | 3 | 1 | 2 | 5 | 6 | 4 |
| 4 | 4 | 5 | 6 | $1+4+5+6$ | $2+4+5+6$ | $3+4+5+6$ |
| 5 | 5 | 6 | 4 | $3+4+5+6$ | $1+4+5+6$ | $2+4+5+6$ |
| 6 | 6 | 4 | 5 | $2+4+5+6$ | $3+4+5+6$ | $1+4+5+6$ |

$$(C.4)$$

The first of these is the $PSU(2)_{10}$ WZW model, discussed already in App. B, with total quantum dimension $\mathcal{D}^2 \approx 44.7846$. This case admits a braiding, but is not modular. The second case corresponds to the Haagerup-Izumi $HI(\mathbb{Z}_3)$ fusion rules, which do not admit a braiding. Unlike the other cases described in this appendix, these fusion rules are non-Abelian. This puts it somewhat outside of the class of fusion rules studied in the main text, though such examples could still have some physical applications, c.f. footnote 10. The total quantum dimension is $\mathcal{D}^2 \approx 35.725$.

For good measure, we also include the nine cases which cannot be realized by fusion categories, leaving it to the reader to come up with useful applications for them,

|   | 1 | 2 | 3 | 4 | 5 | 6 |
|---|---|---|---|---|---|---|
| 1 | 1 | 2 | 3 | 4 | 5 | 6 |
| 2 | 2 | 6 | 1 | 5 | 4 | 3 |
| 3 | 3 | 1 | 6 | 5 | 4 | 2 |
| 4 | 4 | 5 | 5 | $2+3+4+5$ | $1+4+5+6$ | 4 |
| 5 | 5 | 4 | 4 | $1+4+5+6$ | $2+3+4+5$ | 5 |
| 6 | 6 | 3 | 2 | 4 | 5 | 1 |

$$(C.5)$$

|   | 1 | 2 | 3 | 4 | 5 | 6 |
|---|---|---|---|---|---|---|
| 1 | 1 | 2 | 3 | 4 | 5 | 6 |
| 2 | 2 | 6 | 1 | 5 | 4 | 3 |
| 3 | 3 | 1 | 6 | 5 | 4 | 2 |
| 4 | 4 | 5 | 5 | $1+4+5+6$ | $2+3+4+5$ | 4 |
| 5 | 5 | 4 | 4 | $2+3+4+5$ | $1+4+5+6$ | 5 |
| 6 | 6 | 3 | 2 | 4 | 5 | 1 |

$$(C.6)$$

|   | 1 | 2 | 3 | 4 | 5 | 6 |
|---|---|---|---|---|---|---|
| 1 | 1 | 2 | 3 | 4 | 5 | 6 |
| 2 | 2 | $2+3+4+5$ | $1+2+3+6$ | 3 | 3 | 2 |
| 3 | 3 | $1+2+3+6$ | $2+3+4+5$ | 2 | 2 | 3 |
| 4 | 4 | 3 | 2 | 1 | 6 | 5 |
| 5 | 5 | 3 | 2 | 6 | 1 | 4 |
| 6 | 6 | 2 | 3 | 5 | 4 | 1 |

$$(C.7)$$

 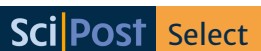

|   | 1 | 2 | 3 | 4 | 5 | 6 |
|---|---|---|---|---|---|---|
| 1 | 1 | 2 | 3 | 4 | 5 | 6 |
| 2 | 2 | 1 | 6 | 5 | 4 | 3 |
| 3 | 3 | 6 | 1 | 5 | 4 | 2 |
| 4 | 4 | 5 | 5 | $1+4+5+6$ | $2+3+4+5$ | 4 |
| 5 | 5 | 4 | 4 | $2+3+4+5$ | $1+4+5+6$ | 5 |
| 6 | 6 | 3 | 2 | 4 | 5 | 1 |

(C.8)

|   | 1 | 2 | 3 | 4 | 5 | 6 |
|---|---|---|---|---|---|---|
| 1 | 1 | 2 | 3 | 4 | 5 | 6 |
| 2 | 2 | 3 | 1 | 5 | 6 | 4 |
| 3 | 3 | 1 | 2 | 6 | 4 | 5 |
| 4 | 4 | 5 | 6 | $3+4+6$ | $1+4+5$ | $2+5+6$ |
| 5 | 5 | 6 | 4 | $1+4+5$ | $2+5+6$ | $3+4+6$ |
| 6 | 6 | 4 | 5 | $2+5+6$ | $3+4+6$ | $1+4+5$ |

(C.9)

|   | 1 | 2 | 3 | 4 | 5 | 6 |
|---|---|---|---|---|---|---|
| 1 | 1 | 2 | 3 | 4 | 5 | 6 |
| 2 | 2 | 1 | 6 | 4 | 5 | 3 |
| 3 | 3 | 6 | $1+4+5+6$ | $3+6$ | $3+6$ | $2+3+4+5$ |
| 4 | 4 | 4 | $3+6$ | $1+2+5$ | $4+5$ | $3+6$ |
| 5 | 5 | 5 | $3+6$ | $4+5$ | $1+2+4$ | $3+6$ |
| 6 | 6 | 3 | $2+3+4+5$ | $3+6$ | $3+6$ | $1+4+5+6$ |

(C.10)

|   | 1 | 2 | 3 | 4 | 5 | 6 |
|---|---|---|---|---|---|---|
| 1 | 1 | 2 | 3 | 4 | 5 | 6 |
| 2 | 2 | 3 | 1 | 5 | 6 | 4 |
| 3 | 3 | 1 | 2 | 6 | 4 | 5 |
| 4 | 4 | 5 | 6 | $3+4+5+6$ | $1+4+5+6$ | $2+4+5+6$ |
| 5 | 5 | 6 | 4 | $1+4+5+6$ | $2+4+5+6$ | $3+4+5+6$ |
| 6 | 6 | 4 | 5 | $2+4+5+6$ | $3+4+5+6$ | $1+4+5+6$ |

(C.11)

|   | 1 | 2 | 3 | 4 | 5 | 6 |
|---|---|---|---|---|---|---|
| 1 | 1 | 2 | 3 | 4 | 5 | 6 |
| 2 | 2 | $2+3+4+5$ | $1+2+3+6$ | 3 | $3+5+6$ | $2+5+6$ |
| 3 | 3 | $1+2+3+6$ | $2+3+4+5$ | 2 | $2+5+6$ | $3+5+6$ |
| 4 | 4 | 3 | 2 | 1 | 6 | 5 |
| 5 | 5 | $3+5+6$ | $2+5+6$ | 6 | $1+2+3$ | $2+3+4$ |
| 6 | 6 | $2+5+6$ | $3+5+6$ | 5 | $2+3+4$ | $1+2+3$ |

(C.12)

|   | 1 | 2 | 3 | 4 | 5 | 6 |
|---|---|---|---|---|---|---|
| 1 | 1 | 2 | 3 | 4 | 5 | 6 |
| 2 | 2 | 1 | 6 | 5 | 4 | 3 |
| 3 | 3 | 6 | $1+4+5$ | $3+5+6$ | $3+4+6$ | $2+4+5$ |
| 4 | 4 | 5 | $3+5+6$ | $1+4+5+6$ | $2+3+4+5$ | $3+4+6$ |
| 5 | 5 | 4 | $3+4+6$ | $2+3+4+5$ | $1+4+5+6$ | $3+5+6$ |
| 6 | 6 | 3 | $2+4+5$ | $3+4+6$ | $3+5+6$ | $1+4+5$ |

(C.13)

Their total quantum dimensions are $\mathcal{D}^2 \approx 18.9282, 18.9282, 18.9282, 18.9282, 20.4853, 25.5826, 35.725, 36.7792,$ and $36.7792$, respectively. In all cases,

$$\mathrm{Com}(A)^2 = \mathrm{Com}(A)^{\infty} = \{1, 2, 3, 4, 5, 6\}\,.$$

## C.4 Rank 7

We finally discuss the case of rank 7, for which there are a total of five multiplicity-1 fusion algebras with conjugate pair length greater than 1 (in particular, 2). Their fusion rules are as follows,

|   | 1 | 2 | 3 | 4 | 5 | 6 | 7 |
|---|---|---|---|---|---|---|---|
| 1 | 1 | 2 | 3 | 4 | 5 | 6 | 7 |
| 2 | 2 | 7 | 1 | 6 | 5 | 4 | 3 |
| 3 | 3 | 1 | 7 | 6 | 5 | 4 | 2 |
| 4 | 4 | 6 | 6 | $1+5+6+7$ | $4+6$ | $2+3+4+5$ | 4 |
| 5 | 5 | 5 | 5 | $4+6$ | $1+2+3+7$ | $4+6$ | 5 |
| 6 | 6 | 4 | 4 | $2+3+4+5$ | $4+6$ | $1+5+6+7$ | 6 |
| 7 | 7 | 3 | 2 | 4 | 5 | 6 | 1 |

$$\tag{C.14}$$

|   | 1 | 2 | 3 | 4 | 5 | 6 | 7 |
|---|---|---|---|---|---|---|---|
| 1 | 1 | 2 | 3 | 4 | 5 | 6 | 7 |
| 2 | 2 | 1 | 7 | 6 | 5 | 4 | 3 |
| 3 | 3 | 7 | 1 | 6 | 5 | 4 | 2 |
| 4 | 4 | 6 | 6 | $1+5+6+7$ | $4+6$ | $2+3+4+5$ | 4 |
| 5 | 5 | 5 | 5 | $4+6$ | $1+2+3+7$ | $4+6$ | 5 |
| 6 | 6 | 4 | 4 | $2+3+4+5$ | $4+6$ | $1+5+6+7$ | 6 |
| 7 | 7 | 3 | 2 | 4 | 5 | 6 | 1 |

$$\tag{C.15}$$

|   | 1 | 2 | 3 | 4 | 5 | 6 | 7 |
|---|---|---|---|---|---|---|---|
| 1 | 1 | 2 | 3 | 4 | 5 | 6 | 7 |
| 2 | 2 | 1 | 7 | 6 | 5 | 4 | 3 |
| 3 | 3 | 7 | $1+6+7$ | $5+7$ | $4+5+6$ | $3+5$ | $2+3+4$ |
| 4 | 4 | 6 | $5+7$ | $1+4+6$ | $3+5+7$ | $2+4+6$ | $3+5$ |
| 5 | 5 | 5 | $4+5+6$ | $3+5+7$ | $1+2+3+4+6+7$ | $3+5+7$ | $4+5+6$ |
| 6 | 6 | 4 | $3+5$ | $2+4+6$ | $3+5+7$ | $1+4+6$ | $5+7$ |
| 7 | 7 | 3 | $2+3+4$ | $3+5$ | $4+5+6$ | $5+7$ | $1+6+7$ |

$$\tag{C.16}$$

|   | 1 | 2 | 3 | 4 | 5 | 6 | 7 |
|---|---|---|---|---|---|---|---|
| 1 | 1 | 2 | 3 | 4 | 5 | 6 | 7 |
| 2 | 2 | 1 | 7 | 6 | 5 | 4 | 3 |
| 3 | 3 | 7 | $1+5+6+7$ | $4+6+7$ | $3+7$ | $3+4+6$ | $2+3+4+5$ |
| 4 | 4 | 6 | $4+6+7$ | $1+3+5+7$ | $4+6$ | $2+3+5+7$ | $3+4+6$ |
| 5 | 5 | 5 | $3+7$ | $4+6$ | $1+2+5$ | $4+6$ | $3+7$ |
| 6 | 6 | 4 | $3+4+6$ | $2+3+5+7$ | $4+6$ | $1+3+5+7$ | $4+6+7$ |
| 7 | 7 | 3 | $2+3+4+5$ | $3+4+6$ | $3+7$ | $4+6+7$ | $1+5+6+7$ |

$$\tag{C.17}$$

|   | 1 | 2 | 3 | 4 | 5 | 6 | 7 |
|---|---|---|---|---|---|---|---|
| 1 | 1 | 2 | 3 | 4 | 5 | 6 | 7 |
| 2 | 2 | 1 | 7 | 4 | 5 | 6 | 3 |
| 3 | 3 | 7 | $1+4+5+6+7$ | $3+7$ | $3+7$ | $3+7$ | $2+3+4+5+6$ |
| 4 | 4 | 4 | $3+7$ | $1+2+6$ | $5+6$ | $4+5$ | $3+7$ |
| 5 | 5 | 5 | $3+7$ | $5+6$ | $1+2+4$ | $4+6$ | $3+7$ |
| 6 | 6 | 6 | $3+7$ | $4+5$ | $4+6$ | $1+2+5$ | $3+7$ |
| 7 | 7 | 3 | $2+3+4+5+6$ | $3+7$ | $3+7$ | $3+7$ | $1+4+5+6+7$ |

$$\tag{C.18}$$

None of these can be realized by a fusion category. The total quantum dimensions are $\mathcal{D}^2 \approx$ 21.1231, 21.1231, 36.9706, 42, and 34.3852, respectively. In all cases $\text{Com}(A)^2 = \text{Com}(A)^\infty = \{1, 2, 3, 4, 5, 6, 7\}$.

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
