# Peer review of "On a class of selection rules without group actions in field theory and string theory"

_SciPost Physics, doi:SciPost Phys. 17, 169 (2024)_

## Round 1 · Referee Report · Anonymous (Referee 2) · 2024-10-18

Report

I thank the authors for addressing all my points in a satisfactory manner. I recommend this submission for publication.

Recommendation

Publish (surpasses expectations and criteria for this Journal; among top 10%)

---

## Round 1 · Referee Report · Anonymous (Referee 1) · 2024-10-18

Report

the authors' updates have addressed all my comments. i recommend this paper for publication.

Recommendation

Publish (surpasses expectations and criteria for this Journal; among top 10%)

---

## Round 1 · Author Response

We thank the two referees for various constructive comments. We made improvements accordingly. In the PDF file of the revised version, the changes are typeset in red, so that they can be easily found. Let us list major changes and our replies to individual points made by the two referees below. We hope these changes and replies will make the manuscript acceptable for publication.

Response to referee report 2:

  1. One important aspect which we should clarify is that the group G appearing in Section 2 is not a symmetry group in the usual sense---indeed, the theory which we discuss (in which fields are labelled by conjugacy of G) does not have a G symmetry. That being said, we do indeed assume that G is finite. It would indeed be interesting to understand the case of infinite G (both discrete and finite), but we relegate this to future work.

  2. To describe the interaction of pointlike charged operators, using a fusion algebra suffices. However, if one wants to consider interactions among both pointlike and extended objects, then one may need to consider higher fusion algebra. We have added a comment to this effect in the revised draft.

  3. As an illustration, we give the multiplication table for the n=2 case (in which case $D_4$ is equivalent to the quaternion group $Q_8$).

Response to referee report 1:

  1. This was common to the point 1 of the referee 2. Please refer to the comments there.

  2. We do not expect tree-level selection rules to accidentally persist at loop-level. This would happen if fields labelled by some elements of the fusion algebra were absent, but we do not consider such "un-faithful" labelings here. We have included a comment on this in the revised draft.

  3. This is an interesting question. Although we do not have any natural example of this type, we added a paragraph showing that there is at least an artificial example for any hypergroup.

  4. We changed the phrasing accordingly.

---

## Editorial Decision

published